# Unlocking the Duality between Flow and Field Matching

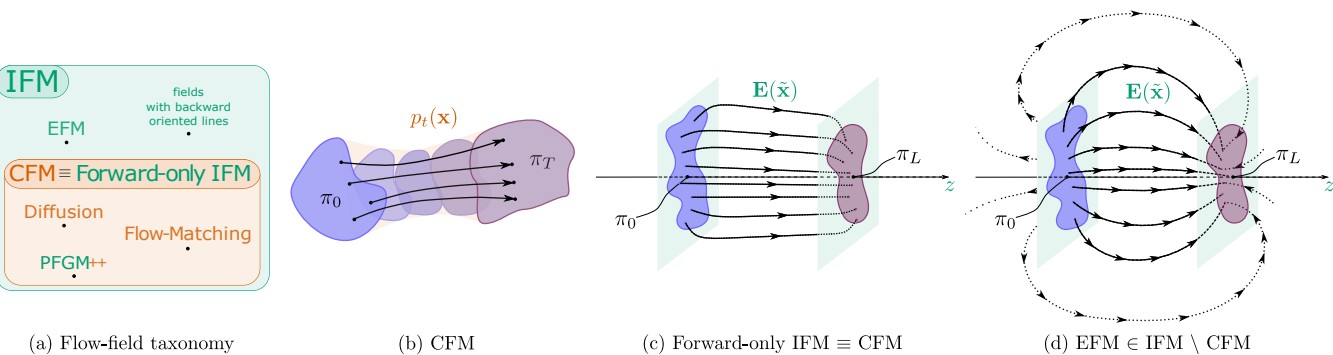

**Figure 1. Our duality overview.** The uncovered duality (§3) unifies two different frameworks, summarized in (a). Conventional CFM (b; §2.1) corresponds exactly to the *forward-only* subclass of IFM (c; §2.2), yielding the same generative dynamics. In contrast, general IFM (§2.2) is strictly more expressive than CFM and can include dyanmics with backward-oriented field lines, such as EFM (d; §2.2).

## Abstract

Conditional Flow Matching (CFM) unifies conventional generative paradigms such as diffusion models and flow matching. Interaction Field Matching (IFM) is a newer framework that generalizes Electrostatic Field Matching (EFM) rooted in Poisson Flow Generative Models (PFGM). While both frameworks define generative dynamics, they start from different objects: CFM specifies a conditional probability path in data space, whereas IFM specifies a physics-inspired interaction field in an augmented data space. This raises a basic question: **are CFM and IFM genuinely different, or are they two descriptions of the same underlying dynamics?** We show that they coincide for a natural subclass of IFM that we call forward-only IFM. Specifically, we construct a bijection between CFM and forward-only IFM. We further show that general IFM is strictly more expressive: it includes EFM and other interaction fields that cannot be realized within the standard CFM formulation. Finally, we highlight how this duality can benefit both frameworks: it provides a probabilistic interpretation of forward-only IFM and yields novel, IFM-driven techniques for CFM.

## 1. Introduction

Diffusion (Sohl-Dickstein et al., 2015; Ho et al., 2020; Song et al., 2020) and flow-based (Liu et al., 2022; Lipman et al., 2022; Albergo & Vanden-Eijnden, 2022) generative models are the dominant approaches in modern generative modeling. These methods can be unified under the Conditional Flow Matching (Tong et al., 2023, CFM) framework, which constructs a generative ODE by first specifying a conditional probability path in data space together with a compatible conditional velocity field, and then marginalizing to obtain the resulting global dynamics that transport the source distribution to the target distribution (Figure 1 (b)).

More recently, a separate line of work has revived physics-inspired *electrostatic* (Xu et al., 2023; Shlenskii & Korotin, 2025) generative modeling, including Poisson Flow Generative Models (Xu et al., 2022a, PFGM) and Electrostatic Field Matching (Kolesov et al., 2025, EFM). These approaches do not naturally fit the CFM formalism. Instead, they can be viewed as instances of the recently proposed field-based Interaction Field Matching (Manukhov et al., 2026, IFM) framework. IFM first specifies a particle-dependent field in an *augmented* space and then applies the superposition principle to obtain a global field whose field lines induce transport between source and target samples (Figure 1).

Although both frameworks define generative dynamics, they are built on different first principles: CFM is path-first and probabilistic, whereas IFM is field-first and geometric. This raises a basic question: *are CFM and IFM fundamentally distinct, or are they the same generative dynamics expressed in different languages?*

[1]Anonymous Institution, Anonymous City, Anonymous Region, Anonymous Country. Correspondence to: Anonymous Author <anon.email@domain.com>.

Preliminary work. Under review by the International Conference on Machine Learning (ICML). Do not distribute.

**Our main contributions** are summarized as follows:

1. **Duality and expressiveness.** We prove that CFM induces the same generative dynamics as a natural subclass of IFM, which we call *forward-only IFM* (§3.2), establishing equivalence at the level of ODE-induced transport. We further show that general IFM is strictly more expressive than CFM (§3.4).

2. **Constructive mappings.** We provide a constructive bijection: mapping conditional flows to forward-only interaction fields (§3.2), and mapping forward-only interaction fields to conditional flows (§3.3). This yields a unified flow–field view of the two frameworks.

3. **Conceptual and practical implications.** We highlight implications of this duality: it endows forward-only IFM with a probabilistic interpretation (§3.4) and suggests new techniques for CFM inspired by IFM (§4).

In summary, we unify conventional CFM and the novel IFM within a single perspective, enabling a coherent development pipeline informed by both flow and field interpretations.

## 2. Preliminaries: CFM vs. IFM?

In this section, we review two frameworks for constructing generative dynamics $d\mathbf{x} = \mathbf{v}_t(\mathbf{x})\,dt$, which transport a source/prior distribution $\pi_T$ or $\pi_L$ (depending on the framework's notation) to a target distribution $\pi_0$. First, in §2.1, we recall the well-established Conditional Flow Matching (Tong et al., 2023, CFM), which generalizes the diffusion and flow-based frameworks (Gao et al., 2025). Next, in §2.2, we review the recently proposed Interaction Field Matching (Manukhov et al., 2026, IFM), which can be viewed as a generalization of electrostatic generative models. In §2.3, we provide a side-by-side comparison of the key ingredients of the two approaches. Finally, in §2.4, we discuss Poisson Flow Generative Models (Xu et al., 2022a, PFGM), which, to our knowledge, is the only framework that admits both flow and field descriptions.

Throughout this section, we use **orange** and **green** coloring to highlight terms specific to CFM and IFM, respectively.

### 2.1. Conditional Flow Matching (CFM)

**Intuition.** The CFM framework begins by specifying **(a)** a **conditional distribution path** $p_t^{\mathbf{x}_0,\mathbf{x}_T}$ that interpolates between $\mathbf{x}_0 \in \mathbb{R}^D$ and $\mathbf{x}_T \in \mathbb{R}^D$, i.e., $p_{t=0}^{\mathbf{x}_0,\mathbf{x}_T} = \delta(\mathbf{x}_0)$ and $p_{t=T}^{\mathbf{x}_0,\mathbf{x}_T} = \delta(\mathbf{x}_T)$; and **(b)** a **conditional velocity** $\mathbf{v}_t^{\mathbf{x}_0,\mathbf{x}_T}(\mathbf{x}) : \mathbb{R}^D \to \mathbb{R}^D$ associated with this conditional path. Both objects are **defined in the data space** $\mathbb{R}^D$ and are conditioned on a target–source pair $\mathbf{x}_0, \mathbf{x}_T$. Marginalizing these conditional objects with respect to a given joint distribution $\pi_{0,T}(\mathbf{x}_0, \mathbf{x}_T)$ yields a velocity field $\mathbf{v}_t$ that induces the generative dynamics $d\mathbf{x} = \mathbf{v}_t(\mathbf{x})\,dt$, transporting the source distribution $\pi_T$ to the target distribution $\pi_0$.

We use blue to emphasize that the elementary CFM objects $p_t^{\mathbf{x}_0,\mathbf{x}_T}(\mathbf{x})$ and $\mathbf{v}_t^{\mathbf{x}_0,\mathbf{x}_T}(\mathbf{x})$ can be constructed both in a *one-sided* or *two-sided* manner. In the one-sided case, $p_t^{\mathbf{x}_0}$ and $\mathbf{v}_t^{\mathbf{x}_0}(\mathbf{x})$ are defined using only target samples $\mathbf{x}_0 \sim \pi_0$. In the two-sided case, $p_t^{\mathbf{x}_0,\mathbf{x}_T}$ and $\mathbf{v}_t^{\mathbf{x}_0,\mathbf{x}_T}(\mathbf{x})$ are defined from paired target and source samples $\mathbf{x}_0, \mathbf{x}_T \sim \pi_{0,T}$, for example under the independent coupling $\pi_{0,T} = \pi_0 \times \pi_T$. This substitution does not affect the theoretical guarantees (Tong et al., 2023). The one-sided setting is common in noise-to-data generation (Ho et al., 2020), where the source distribution $\pi_T$ is a simple prior (e.g., a standard Gaussian) independent of the target distribution $\pi_0$, whereas the two-sided setting is more typical for data-to-data translation (Liu et al., 2022; Albergo et al., 2023).

**Definition (Conditional Flow).** A *Conditional Flow* (Tong et al., 2023) is specified by a pair $(\mathbf{v}_t^{\mathbf{x}_0,\mathbf{x}_T}(\mathbf{x}), p_t^{\mathbf{x}_0,\mathbf{x}_T}(\mathbf{x}))$ consisting of a conditional velocity field and a path of conditional distributions. Both objects depend on $\mathbf{x}_0, \mathbf{x}_T$ and satisfy the ***continuity equation*** $\forall \mathbf{x} \in \mathbb{R}^D$ and $t \in (0,T)$

$$\partial_t p_t^{\mathbf{x}_0,\mathbf{x}_T}(\mathbf{x}) + \nabla_{\mathbf{x}} \cdot \left(\mathbf{v}_t^{\mathbf{x}_0,\mathbf{x}_T}(\mathbf{x})\, p_t^{\mathbf{x}_0,\mathbf{x}_T}(\mathbf{x})\right) = 0, \quad (1)$$

where $\nabla_{\mathbf{x}}\cdot$ denotes the divergence operator with respect to $\mathbf{x}$, together with the *boundary conditions*

$$\lim_{t \to 0} p_t^{\mathbf{x}_0,\mathbf{x}_T}(\mathbf{x}) = \delta_{\mathbf{x}_0}(\mathbf{x}), \quad \lim_{t \to T} p_t^{\mathbf{x}_0,\mathbf{x}_T}(\mathbf{x}) = \delta_{\mathbf{x}_T}(\mathbf{x}), \quad (2)$$

that hold in the sense of weak convergence.

**Generative dynamics.** This pointwise-defined Conditional Flow induces a generative model that transports a source distribution $\pi_T(\mathbf{x})$ to a target distribution $\pi_0(\mathbf{x})$. First, marginalizing the conditional path with respect to the data distribution $\pi_{0,T}$ yields a **path of marginal distributions**

$$p_t(\mathbf{x}) = \int p_t^{\mathbf{x}_0,\mathbf{x}_T}(\mathbf{x})\, \pi_{0,T}(\mathbf{x}_0, \mathbf{x}_T)\, d\mathbf{x}_0\, d\mathbf{x}_T, \quad (3)$$

which satisfies $p_{t=0} = \pi_0$ and $p_{t=T} = \pi_T$. Second, one can show that the marginals $p_t$ are generated by a deterministic flow governed by the ODE

$$\frac{d\mathbf{x}}{dt} = \mathbf{v}_t(\mathbf{x}) = \mathbb{E}_{p_t(\mathbf{x}_0,\mathbf{x}_T|\mathbf{x}_t=\mathbf{x})} \mathbf{v}_t^{\mathbf{x}_0,\mathbf{x}_T}(\mathbf{x}), \quad (4)$$

where the **global velocity** $\mathbf{v}_t(\mathbf{x})$ is the conditional expectation of the local velocity $\mathbf{v}_t^{\mathbf{x}_0,\mathbf{x}_T}(\mathbf{x})$ given that $\mathbf{x}_t \sim p_t^{\mathbf{x}_0,\mathbf{x}_T}$.

Although the drift expression (4) is explicit, it is generally intractable to evaluate in practice, since sampling from the conditional distribution $p_t(\mathbf{x}_0, \mathbf{x}_T \mid \mathbf{x}_t)$ is infeasible.

**Training.** The conditional expectation in (4) naturally yields a regression objective for estimating the drift. In particular, $\mathbf{v}_t(\mathbf{x})$ can be learned by minimizing

$$\mathbb{E}_{t,\mathbf{x}_0,\mathbf{x}_T,\mathbf{x}_t} \left\|\mathbf{v}_t(\mathbf{x}_t) - \mathbf{v}_t^{\mathbf{x}_0,\mathbf{x}_T}(\mathbf{x}_t)\right\|_2^2, \quad (5)$$

where $t \sim \mathcal{U}[0,T]$ and $\mathbf{x}_t \sim p_t^{\mathbf{x}_0,\mathbf{x}_T}$, and the conditional velocity $\mathbf{v}_t^{\mathbf{x}_0,\mathbf{x}_T}$ provides a **single-sample target**.

**Instantiation I (Two-sided Stochastic Interpolant).** We

now describe two commonly used instances of the Conditional Flow Matching framework. The first is the two-sided Stochastic Interpolant (Albergo et al., 2023, §2.1), defined by $\mathbf{x}_t = \mathcal{I}(t, \mathbf{x}_0, \mathbf{x}_T) + \sigma(t)\,\epsilon$, where **(a)** $\mathcal{I}(0, \mathbf{x}_0, \mathbf{x}_T) = \mathbf{x}_0$ and $\mathcal{I}(T, \mathbf{x}_0, \mathbf{x}_T) = \mathbf{x}_T$; **(b)** $\sigma(0) = 0$, $\sigma(T) = 0$, and $\sigma(t) > 0$ for all $t \in (0, T)$; **(c)** $\epsilon \sim \mathcal{N}(0, I)$ is independent of $\mathbf{x}_0, \mathbf{x}_T$. This construction induces a path of conditional distributions $p_t^{\mathbf{x}_0, \mathbf{x}_T}(\mathbf{x}) = \mathcal{N}\big(\mathcal{I}(t, \mathbf{x}_0, \mathbf{x}_T), \sigma(t)^2 I\big)$, with the associated conditional velocity field $\mathbf{v}_t^{\mathbf{x}_0, \mathbf{x}_T}(\mathbf{x}_t) = \dot{\mathcal{I}}(t, \mathbf{x}_0, \mathbf{x}_T) + \frac{\dot{\sigma}(t)}{\sigma(t)}\big(\mathbf{x}_t - \mathcal{I}(t, \mathbf{x}_0, \mathbf{x}_T)\big)$, where a dot over a function denotes the partial derivative with respect to time.

**Instantiation II (One-sided Stochastic Interpolant).** The second instance is the one-sided Stochastic Interpolant (Albergo et al., 2023, §3.2), also known as Flow Matching (Liu et al., 2022; Lipman et al., 2022), defined by $\mathbf{x}_t = J(t, \mathbf{x}_0) + \sigma(t)\,\epsilon$, where **(a)** $J(0, \mathbf{x}_0) = \mathbf{x}_0$ and $J(T, \mathbf{x}_0) = \mathbf{0}$; **(b)** $\sigma(0) = 0$, $\sigma(T) = 1$, and $\sigma(t) > 0$ for all $t \in (0, T)$; **(c)** $\mathbf{x}_T \sim \mathcal{N}(0, I)$ is independent of $\mathbf{x}_0$. As in the two-sided case, this yields a path of conditional distributions $p_t^{\mathbf{x}_0}(\mathbf{x}) = \mathcal{N}\big(J(t, \mathbf{x}_0), \sigma(t)^2 I\big)$, with the corresponding conditional velocity field $\mathbf{v}_t^{\mathbf{x}_0}(\mathbf{x}_t) = \dot{J}(t, \mathbf{x}_0) + \frac{\dot{\sigma}(t)}{\sigma(t)}\big(\mathbf{x}_t - J(t, \mathbf{x}_0)\big)$.

*Remark.* Although this construction conditions only on $\mathbf{x}_0$, it still specifies a right-endpoint prior $\pi_T$ that is independent of $\pi_0$, namely the standard Gaussian: $\pi_T(\mathbf{x}_T \mid \mathbf{x}_0) = \pi_T(\mathbf{x}_T) = \mathcal{N}(0, I)$. Moreover, in the special case $J(t, \mathbf{x}_0) = \mathbf{x}_0(1 - t/T)$ and $\sigma(t) = t/T$, one recovers the classical linear interpolant introduced in (Lipman et al., 2022). The same choice also recovers the corresponding formulation for diffusion models (Gao et al., 2025).

## 2.2. Interaction Field Matching (IFM)

**Intuition.** The IFM framework operates in an **extended space** $\mathbb{R}^{D+1}$, where each data point $\mathbf{x} \in \mathbb{R}^D$ is augmented with an additional coordinate $z \in \mathbb{R}$ to form $\tilde{\mathbf{x}} = (\mathbf{x}, z)$. The target and source distributions are placed on two separate *plates* located at $z = 0$ and $z = L > 0$, respectively. Accordingly, samples $\mathbf{x}_0 \sim \pi_0$ and $\mathbf{x}_L \sim \pi_L$ correspond to $\tilde{\mathbf{x}}_0 = (\mathbf{x}_0, 0)$ and $\tilde{\mathbf{x}}_L = (\mathbf{x}_L, L)$. In Electrostatic Field Matching (Kolesov et al., 2025, EFM), this geometric setup is interpreted as a $(D+1)$-dimensional *electric capacitor*.

Within this construction, IFM begins by introducing an **interaction field** $\mathbf{E}^{\mathbf{x}_0, \mathbf{x}_L}(\tilde{\mathbf{x}}) : \mathbb{R}^{D+1} \rightarrow \mathbb{R}^{D+1}$, defined for a pair of points $\mathbf{x}_0, \mathbf{x}_L$, whose **field lines** connect $\tilde{\mathbf{x}}_0$ and $\tilde{\mathbf{x}}_L$. A corresponding **global field** $\mathbf{E}(\tilde{\mathbf{x}})$ is then obtained via the **superposition principle** by averaging these interaction fields over a given joint distribution $\pi_{0,L}$. The field lines of the resulting global field define the transport from the source distribution $\pi_L$ to the target distribution $\pi_0$.

*Remark.* In contrast to Conditional Flows, Interaction Fields are inherently two-sided: their field lines are required to connect two particles located at $\tilde{\mathbf{x}}_0$ and $\tilde{\mathbf{x}}_L$. This is reflected in our notation: for Conditional Flows, the source conditioning point $\mathbf{x}_T$ is shown in blue, whereas for Interaction Fields the source point $\mathbf{x}_L$ is treated on the same basis as the target point $\mathbf{x}_0$ and is therefore shown in black.

**Definition (Interaction Field).** A vector field $\mathbf{E}^{\mathbf{x}_0, \mathbf{x}_L}(\tilde{\mathbf{x}}) \in \mathbb{R}^{D+1}$ is an *Interaction Field* (Manukhov et al., 2026) if it satisfies the following three properties.

First, its **field lines** *connect the two particles*:

$$\begin{cases} \dfrac{d\tilde{\mathbf{x}}(\tau)}{d\tau} = \mathbf{E}^{\mathbf{x}_0, \mathbf{x}_L}(\tilde{\mathbf{x}}(\tau)), \\ \tilde{\mathbf{x}}(\tau_s) = \tilde{\mathbf{x}}_0, \quad \tilde{\mathbf{x}}(\tau_f) = \tilde{\mathbf{x}}_L, \end{cases} \quad (6)$$

where $\tau_s$ and $\tau_f$ denote the start and end values of the field-line parameter, respectively.

*Remark.* The parameter $\tau$ in (6) is not a global schedule like the time variable in Conditional Flows (1). Rather, $\tau$ is an arbitrary scalar parameter used solely to trace a field line (integral curve) of $\mathbf{E}^{\mathbf{x}_0, \mathbf{x}_L}(\tilde{\mathbf{x}})$.

Second, the field is **divergence-free (flux-conserving)**:

$$\nabla \cdot \mathbf{E}^{\mathbf{x}_0, \mathbf{x}_L}(\tilde{\mathbf{x}}) = 0. \quad (7)$$

Third, the field has a **fixed total flux**: for any closed surface $\partial M$ enclosing $\tilde{\mathbf{x}}_0$ but not $\tilde{\mathbf{x}}_L$, the total flux

$$\int_{\partial M} \mathbf{E}^{\mathbf{x}_0, \mathbf{x}_L}(\tilde{\mathbf{x}}) \cdot d\mathbf{S} = \Phi_0 \quad (8)$$

is independent of the particle locations $\tilde{\mathbf{x}}_0, \tilde{\mathbf{x}}_L$.

**Special case (forward-only Interaction Field).** We say that a field defined on the region between the plates ($0 \le z \le L$) is *forward-only* if its $z$-component satisfies $\mathbf{E}(\tilde{\mathbf{x}})_z > 0$ everywhere the field is nonzero. An Interaction Field with this property is called a *forward-only Interaction Field*, which is the main focus of this work.

**Generative dynamics.** Given the target and source distributions $\pi_0$ and $\pi_L$, the induced **global field** $\mathbf{E}(\tilde{\mathbf{x}})$ is defined via a **generalized superposition principle** as

$$\mathbf{E}(\tilde{\mathbf{x}}) = \int \mathbf{E}^{\mathbf{x}_0, \mathbf{x}_L}(\tilde{\mathbf{x}})\, \pi_{0,L}(\mathbf{x}_0, \mathbf{x}_L)\, d\mathbf{x}_0\, d\mathbf{x}_L. \quad (9)$$

That is, $\mathbf{E}(\tilde{\mathbf{x}})$ is the average contribution of interaction fields over particle pairs $\mathbf{x}_0, \mathbf{x}_L \sim \pi_{0,L}$. As shown in Manukhov et al. (2026), one can transport samples from $\pi_0$ to $\pi_L$ by following the field lines of $\mathbf{E}$:

$$\frac{d\tilde{\mathbf{x}}(\tau)}{d\tau} = \mathbf{E}(\tilde{\mathbf{x}}(\tau)). \quad (10)$$

In general, simulating (10) poses practical difficulties: **(a)** the boundary conditions are implicit because the corresponding $\tau_s$ and $\tau_f$ are unknown; **(b)** field lines may move both forward ($\mathbf{E}(\tilde{\mathbf{x}})_z > 0$) and backward ($\mathbf{E}(\tilde{\mathbf{x}})_z < 0$) along the $z$-axis; and **(c)** field lines may be defined beyond the

*Table 1.* **CFM vs. IFM**: side-by-side comparison of the Conditional Flow Matching and Interaction Field Matching frameworks.

| Aspect | Conditional Flow Matching | Interaction Field Matching |
|---|---|---|
| **Elementary object** | **Conditional Flow** (§2.1) $(\mathbf{v}_t^{\mathbf{x}_0,\mathbf{x}_T}(\mathbf{x}), \, p_t^{\mathbf{x}_0,\mathbf{x}_T}(\mathbf{x}))$ defined **in the data space** $\mathbb{R}^D$ | **Interaction Field** (6) $\mathbf{E}^{\mathbf{x}_0,\mathbf{x}_L}(\tilde{\mathbf{x}})$ defined **in the extended space** $\mathbb{R}^{D+1}$ |
| **Global object** | **Conditional averaging** of conditional velocities: $\mathbf{v}_t(\mathbf{x}) = \int \mathbf{v}_t^{\mathbf{x}_0,\mathbf{x}_T}(\mathbf{x}) p_t(\mathbf{x}_0, \mathbf{x}_T \mid \mathbf{x}_t = \mathbf{x}) d\mathbf{x}_0 d\mathbf{x}_T$ $\neq \int \mathbf{v}_t^{\mathbf{x}_0,\mathbf{x}_T}(\mathbf{x}) \, \pi_{0,T}(\mathbf{x}_0, \mathbf{x}_T) \, d\mathbf{x}_0 \, d\mathbf{x}_T$ | **Generalized superposition** of pairwise fields: $\mathbf{E}(\tilde{\mathbf{x}}) = \int \mathbf{E}^{\mathbf{x}_0,\mathbf{x}_L}(\tilde{\mathbf{x}}) \, \pi_{0,L}(\mathbf{x}_0, \mathbf{x}_L) \, d\mathbf{x}_0 \, d\mathbf{x}_L$ |
| **Generative dynamics** | ODE-based dynamics **in the data space**: $\dfrac{d\mathbf{x}}{dt} = \mathbf{v}_t(\mathbf{x}), \quad \mathbf{x}_0 \sim \pi_0$ | Field-line transport **in the extended space**: $\dfrac{d\tilde{\mathbf{x}}}{d\tau} = \mathbf{E}(\tilde{\mathbf{x}}), \quad \tilde{\mathbf{x}} \sim \pi_0 \times \delta_{z=0}$ **Forward-only** fields dynamics **in the data space**: $\dfrac{d\mathbf{x}}{dz} = \dfrac{\mathbf{E}(\tilde{\mathbf{x}})_{\mathbf{x}}}{\mathbf{E}(\tilde{\mathbf{x}})_z}, \quad \mathbf{x}_0 \sim \pi_0$ |
| **Marginal distributions** | **Explicitly** defined via conditional distributions: $p_t(\mathbf{x}) = \int p_t^{\mathbf{x}_0,\mathbf{x}_T}(\mathbf{x}) \, \pi_{0,T}(\mathbf{x}_0, \mathbf{x}_T) \, d\mathbf{x}_0 \, d\mathbf{x}_T$ | **No explicit** marginal density path: defined implicitly via field-line transport (10) |
| **Training objective** | **Single-sample conditional velocity** regression: $\mathbb{E}_{t, \mathbf{x}_0, \mathbf{x}_T, \mathbf{x}_t} \left\| \mathbf{v}_t(\mathbf{x}_t) - \mathbf{v}_t^{\mathbf{x}_0,\mathbf{x}_T}(\mathbf{x}_t) \right\|_2^2$ | **Multi-sample global field** regression: $\mathbb{E}_{\tilde{\mathbf{x}} \sim p_{\mathrm{vol}}} \left\| f_\theta(\tilde{\mathbf{x}}) - \mathbf{E}(\tilde{\mathbf{x}}) \right\|_2^2$ **Normalized field** in the forward-only case: $\mathbb{E}_{\tilde{\mathbf{x}} \sim p_{\mathrm{vol}}} \left\| f_\theta(\tilde{\mathbf{x}}) - \dfrac{\mathbf{E}(\tilde{\mathbf{x}})}{\|\mathbf{E}(\tilde{\mathbf{x}})\|_2} \right\|_2^2$ |
| **Training volume** | **Explicitly** defined by conditional distributions: $\mathbf{x}_0, \mathbf{x}_T \sim \pi_{0,T}, \ t \sim \mathcal{U}[0,T], \ \mathbf{x}_t \sim p_t^{\mathbf{x}_0,\mathbf{x}_T}$ | **Ad hoc heuristic** distribution: $\tilde{\mathbf{x}} \sim p_{\mathrm{vol}}$ |

capacitor region, $0 \leq z \leq L$.

For a forward-only Interaction Field, the global field is forward-only as well, i.e., $\mathbf{E}_z > 0$, and (10) can be reparameterized using the physically meaningful coordinate $z$:

$$\frac{d\tilde{\mathbf{x}}}{dz} = \frac{d\tilde{\mathbf{x}}}{d\tau} \cdot \frac{d\tau}{dz} = \mathbf{E}(\tilde{\mathbf{x}}) \, \mathbf{E}(\tilde{\mathbf{x}})_z^{-1} = \left( \frac{\mathbf{E}(\tilde{\mathbf{x}})_{\mathbf{x}}}{\mathbf{E}(\tilde{\mathbf{x}})_z}, 1 \right). \quad (11)$$

This form yields explicit boundaries, $z(\tau_s) = 0$ and $z(\tau_f) = L$, and induces generative dynamics in the data space $\mathbb{R}^D$:

$$\frac{d\mathbf{x}}{dz} = \frac{\mathbf{E}(\tilde{\mathbf{x}})_{\mathbf{x}}}{\mathbf{E}(\tilde{\mathbf{x}})_z}, \quad (12)$$

where $z$ plays the role of a time-like variable.

**Training.** For a general Interaction Field, one has to estimate the full field $\mathbf{E}(\tilde{\mathbf{x}}) \in \mathbb{R}^{D+1}$ at every point $\tilde{\mathbf{x}} \in \mathbb{R}^{D+1}$ where the field is defined. For forward-only fields, however, it suffices to estimate the *normalized* field $\mathbf{E}(\tilde{\mathbf{x}})/\|\mathbf{E}(\tilde{\mathbf{x}})\|_2$. Indeed, the generative dynamics (12) can be rewritten as

$$\frac{d\mathbf{x}}{dz} = \frac{\mathbf{E}(\tilde{\mathbf{x}})_{\mathbf{x}}}{\mathbf{E}(\tilde{\mathbf{x}})_z} = \frac{\mathbf{E}(\tilde{\mathbf{x}})_{\mathbf{x}}}{\|\mathbf{E}(\tilde{\mathbf{x}})\|_2} \cdot \frac{\|\mathbf{E}(\tilde{\mathbf{x}})\|_2}{\mathbf{E}(\tilde{\mathbf{x}})_z}. \quad (13)$$

Motivated by this observation, Manukhov et al. (2026) use the following training objective:

$$\mathbb{E}_{\tilde{\mathbf{x}} \sim p_{\mathrm{vol}}} \left\| f_\theta(\tilde{\mathbf{x}}) - \frac{\mathbf{E}(\tilde{\mathbf{x}})}{\|\mathbf{E}(\tilde{\mathbf{x}})\|_2} \right\|_2^2, \quad (14)$$

where **(a)** $f_\theta : \mathbb{R}^{D+1} \to \mathbb{R}^{D+1}$ is a neural network that estimates the normalized Interaction Field; **(b)** $p_{\mathrm{vol}}$ is a heuristic **distribution intended to cover the volume** between the plates at $z = 0$ and $z = L$, where the field is defined; and **(c)** $\mathbf{E}(\tilde{\mathbf{x}})$ is the global field estimated from data samples according to the generalized superposition principle (9).

**Instantiation I (Electrostatic Field).** An early example of an Interaction Field is provided by the Electrostatic Field Matching (Kolesov et al., 2025, EFM) framework. Consider a unit point charge of sign $\pm 1$ located at $\tilde{\mathbf{x}}' \in \mathbb{R}^{D+1}$. By Coulomb's law, it induces the field $\mathbf{E}_{\pm}^{\tilde{\mathbf{x}}'}(\tilde{\mathbf{x}}) \propto \pm(\tilde{\mathbf{x}} - \tilde{\mathbf{x}}')/\|\tilde{\mathbf{x}} - \tilde{\mathbf{x}}'\|_2^{D+1}$. Assigning a positive unit charge to augmented samples $\tilde{\mathbf{x}}_0 \sim \pi_0 \times \delta_{z=0}$ and a negative to $\tilde{\mathbf{x}}_L \sim \pi_L \times \delta_{z=L}$ yields, by the electrostatic superposition principle (Xu et al., 2022b), the pairwise-defined field $\mathbf{E}^{\mathbf{x}_0,\mathbf{x}_L}(\tilde{\mathbf{x}}) = \mathbf{E}_+^{\tilde{\mathbf{x}}_0}(\tilde{\mathbf{x}}) + \mathbf{E}_-^{\tilde{\mathbf{x}}_L}(\tilde{\mathbf{x}})$. This field

is an Interaction Field in the sense of §2.2. Moreover, it is nonzero on the entire space $\mathbb{R}^{D+1}$, and its field lines may move backward along the $z$-axis (Manukhov et al., 2026, §2.3). Therefore, the EFM field is not forward-only.

**Instantiation II (IFM Field Realization).** As discussed above, fields with backward-oriented lines are not straightforward to use in practice. To address this, IFM (Manukhov et al., 2026) proposes a forward-only realization. They decompose the field as $\mathbf{E}^{\mathbf{x}_0,\mathbf{x}_L}(\tilde{\mathbf{x}}) = \|\mathbf{E}^{\mathbf{x}_0,\mathbf{x}_L}(\tilde{\mathbf{x}})\|_2 \, \mathbf{n}^{\mathbf{x}_0,\mathbf{x}_L}(\tilde{\mathbf{x}})$, where $\tilde{\mathbf{x}} = (\mathbf{x}, z)$. In (Manukhov et al., 2026, Appendix A.4), they give a concrete procedure to specify $\|\mathbf{E}^{\mathbf{x}_0,\mathbf{x}_L}(\tilde{\mathbf{x}})\|_2$ and $\mathbf{n}^{\mathbf{x}_0,\mathbf{x}_L}(\tilde{\mathbf{x}})$, which we reproduce in Appendix A, for convenience.

## 2.3. CFM vs. IFM

Previously, we reviewed the CFM and IFM frameworks separately. Although their differences may already be apparent, we include Table 1 for convenience; it lists the main terms of both frameworks side by side.

The key differences can be summarized as follows. First, the IFM **global field** can be estimated directly from samples via the **superposition principle** (9), whereas the CFM **global drift**, defined through a **conditional expectation** (4), is generally intractable. Second, the training procedures differ: CFM uses a **single-sample conditional velocity as the target** (32) and learns it at points drawn from the conditional distribution, while IFM uses the **global drift as the target** (14) and learns it at points drawn from an ad hoc **volume coverage distribution**.

Despite these differences, the following example (§2.4) suggests that they actually may have much in common.

## 2.4. Poisson Flow Generative Models

Poisson Flow Generative Models (Xu et al., 2022a, PFGM) is a noise-to-data generative framework rooted in electrostatics (akin to EFM). It is of particular interest here because it admits both a field-based and a flow-based interpretation. To our knowledge, PFGM is the only existing framework that supports both viewpoints.

**Field-based interpretation.** PFGM embeds the data distribution into an extended space $\mathbb{R}^{D+1}$ by placing each sample $\mathbf{x}_0 \sim \pi_0$ on the plane $\{z = 0\}$, i.e., at $\tilde{\mathbf{x}}_0 = (\mathbf{x}_0, 0)$. Each $\tilde{\mathbf{x}}_0$ is treated as a unit positive charge, inducing an electric field in accordance with Coulomb's law:

$$\mathbf{E}^{\mathbf{x}_0}(\tilde{\mathbf{x}}) \propto \frac{\tilde{\mathbf{x}} - \tilde{\mathbf{x}}_0}{\|\tilde{\mathbf{x}} - \tilde{\mathbf{x}}_0\|_2^{D+1}}, \tag{15}$$

for $\tilde{\mathbf{x}} = (\mathbf{x}, z) \in \mathbb{R}^{D+1}$. The global field is obtained via the *electrostatic superposition principle* as $\mathbf{E}(\tilde{\mathbf{x}}) = \int \mathbf{E}^{\mathbf{x}_0}(\tilde{\mathbf{x}}) \, \pi_0(\mathbf{x}_0) \, d\mathbf{x}_0$. The field lines of $\mathbf{E}$ then define a transport from the data distribution to a prior $\pi_L$, which is

a uniform distribution over a hemisphere of radius $L$ (with $L \to \infty$ or sufficiently large in practice).

*Remark.* Despite its resemblance to IFM, PFGM is not a concrete IFM instance: its conditional field is one-sided (constructed from a single target point $\mathbf{x}_0$), whereas IFM requires two-sided conditioning.

**Flow-based interpretation.** Subsequent work (Xu et al., 2023, PFGM++) reveals that PFGM possesses an underlying flow-based structure. Specifically, it demonstrates that PFGM's data-space generative dynamics, as in (12), can be obtained by minimizing the objective

$$\mathbb{E}_{z,\mathbf{x}_0,\mathbf{x}_z} \left\| f_\theta(\mathbf{x}_z, z) - \frac{\mathbf{x}_z - \mathbf{x}_0}{z} \right\|_2^2, \tag{16}$$

where $z \sim \mathcal{U}[0, L]$ and $\mathbf{x}_z \sim p_z^{\mathbf{x}_0}$, with

$$p_z^{\mathbf{x}_0}(\mathbf{x}) \propto \left( \|\mathbf{x} - \mathbf{x}_0\|_2^2 + z^2 \right)^{-\frac{D+1}{2}}. \tag{17}$$

The similarity between (16) and the CFM objective (32) suggests a flow-based interpretation of PFGM: (17) acts as a conditional path distribution, and $(\mathbf{x}_z - \mathbf{x}_0)/z$ serves as the corresponding conditional velocity target.

> **Summary.** PFGM example shows that a single generative transport can be described either as a *flow* or as *field lines*. This motivates two questions:
>
> (1) Is there a general CFM–IFM duality? If so, what can we transfer between the two viewpoints?
>
> (2) Can IFM admit a one-sided (target-only) formulation, analogous to one-sided CFM?

## 3. Duality of CFM and forward-only IFM

In this section, we formally establish the duality between CFM and forward-only IFM. We begin in §3.1 by introducing a one-sided formulation of IFM, which allows us to prove the duality in full generality, covering both the one-sided and two-sided settings. Then, in §3.2, we show how forward-only IFM dynamics can be derived directly from CFM dynamics. Conversely, in §3.3, we demonstrate that CFM dynamics can be recovered from forward-only IFM, completing the proof of duality. Finally, in §3.4, we discuss key theoretical insights arising from this equivalence. All proofs are provided in Appendix B.

Before stating our main results, we make a simple observation. In forward-only fields, the spatial coordinate $z$ (the distance from the left capacitor plate) plays the same role as the time variable $t$ in Conditional Flows. Likewise, the plate separation $L$ corresponds to the terminal time $T$. With this correspondence in mind, we will use the $(t, T)$-based notation for both Conditional Flows and forward-only Interaction Fields throughout the remainder of the paper.

### 3.1. One-sided IFM

Standard IFM is defined in a two-sided form. To state our duality in full generality, we begin by introducing a one-sided notion of an interaction field. Following PFGM (§2.4) and the general IFM formulation (§2.2), we define the one-sided interaction field on the extended space $\mathbb{R}^{D+1}$, where the data distribution lies on a hyperplane $\{t = 0\}$.

**Definition (One-sided Interaction Field).** A vector field $\mathbf{E}^{\mathbf{x}_0}(\tilde{\mathbf{x}}) \in \mathbb{R}^{D+1}$ is called a one-sided interaction field if it satisfies the following three properties.

First, its field lines start at the particle $\tilde{\mathbf{x}}_0 = (\mathbf{x}_0, 0)$:

$$\frac{d\tilde{\mathbf{x}}(\tau)}{d\tau} = \mathbf{E}^{\mathbf{x}_0}(\tilde{\mathbf{x}}(\tau)), \quad \tilde{\mathbf{x}}(\tau_s) = \tilde{\mathbf{x}}_0 \quad (18)$$

where $\tau_s$ denotes the starting value of the field-line parameter and ends anywhere at hyperplane $\{t = T\}$.

Second, the field is *divergence-free (flux-conserving)*:

$$\nabla \cdot \mathbf{E}^{\mathbf{x}_0}(\tilde{\mathbf{x}}) = 0. \quad (19)$$

Third, the field has a *fixed total flux*: for any closed surface $\partial M$ that encloses the particle at $\tilde{\mathbf{x}}_0$, the total flux

$$\int_{\partial M} \mathbf{E}^{\mathbf{x}_0}(\tilde{\mathbf{x}}) \cdot d\mathbf{S} = \Phi_0 \quad (20)$$

is independent of the particle location $\tilde{\mathbf{x}}_0$.

*Remark.* This definition matches the two-sided Interaction Field, except that the field lines are not required to end at any predetermined particle.

**Generative dynamics.** The global field induced by the distribution $\pi_0$ is obtained via generalized superposition, analogously to (9):

$$\mathbf{E}(\tilde{\mathbf{x}}) = \int \mathbf{E}^{\mathbf{x}_0}(\tilde{\mathbf{x}}) \, \pi_0(\mathbf{x}_0) \, d\mathbf{x}_0. \quad (21)$$

The corresponding field lines (10) push forward $\pi_0$ to a distribution $\pi_T$. Consequently, one-sided IFM defines a source (prior) distribution $\pi_T$ and transports between $\pi_T$ and $\pi_0$ along the global field lines.

*Remark.* In contrast to two-sided IFM, the generative dynamics in one-sided IFM is essentially built into the construction, since the source (prior) distribution $\pi_T$ is defined as the pushforward induced by the field lines. In the two-sided setting, both endpoint distributions are specified in advance, and the fact that the field lines induce transport between them is a nontrivial property.

**Example (PFGM $\in$ forward-only one-sided IFM).** Indeed, one can show that the PFGM field defined in §2.4 is an instance of one-sided IFM Moreover, it is foward-only field meaning $\mathbf{E}^{\mathbf{x}_0}(\tilde{\mathbf{x}})_t > 0$ for any $t > 0$.

Given the introduced definition, as in CFM, we use blue to indicate that the source-conditioning point in the IFM field

can be omitted, yielding the one-sided case.

### 3.2. From Flow to Fields

We first establish the flow-to-field direction of the duality. This is formalized in our following theorem.

**Theorem 3.1.** (CFM $\subseteq$ forward-only IFM). *Let $\pi_0$ and $\pi_T$ be $D$-dimensional target and source distributions, respectively. Let $(\mathbf{v}_t^{\mathbf{x}_0,\mathbf{x}_T}(\mathbf{x}), p_t^{\mathbf{x}_0,\mathbf{x}_T}(\mathbf{x}))$ be a Conditional Flow (§2.1), and let $\mathbf{v}_t(\mathbf{x})$ and $p_t(\mathbf{x})$ denote the global velocity (4) and marginal probability path (3) obtained by marginalizing over $\pi_{0,T}(\mathbf{x}_0, \mathbf{x}_T)$, respectively. Define the $(D+1)$-dimensional vector field*

$$\mathbf{E}^{\mathbf{x}_0,\mathbf{x}_T}(\mathbf{x}, t) := (\underbrace{\mathbf{v}_t^{\mathbf{x}_0,\mathbf{x}_T}(\mathbf{x}) \, p_t^{\mathbf{x}_0,\mathbf{x}_T}(\mathbf{x})}_{\mathbf{E}^{\mathbf{x}_0,\mathbf{x}_T}(\mathbf{x},t)_{\mathbf{x}}}, \underbrace{p_t^{\mathbf{x}_0,\mathbf{x}_T}(\mathbf{x})}_{\mathbf{E}^{\mathbf{x}_0,\mathbf{x}_T}(\mathbf{x},t)_t}). \quad (22)$$

*Then, under mild assumptions specified in Appendix B.1:*

1. *The vector field $\mathbf{E}^{\mathbf{x}_0,\mathbf{x}_T}$ is a forward-only Interaction Field (§2.2) with a unit total flux $\Phi_0 = 1$.*

2. *The global field $\mathbf{E}(\mathbf{x}, t)$ obtained via the generalized superposition principle (9) can be recovered as*

$$\mathbf{E}(\mathbf{x}, t) = (\mathbf{v}_t(\mathbf{x}) \, p_t(\mathbf{x}), \, p_t(\mathbf{x})). \quad (23)$$

3. *The CFM generative dynamics (4) coincide with the dynamics induced by field lines (10), reparameterized by the distance from the plate (12):*

$$\frac{d\mathbf{x}}{dt} = \mathbf{v}_t(\mathbf{x}) = \frac{\mathbf{E}(\mathbf{x}, t)_{\mathbf{x}}}{\mathbf{E}(\mathbf{x}, t)_t}. \quad (24)$$

Our Theorem 3.1 shows that any Conditional Flow can be represented as a forward-only Interaction Field that induces the same generative dynamics. Moreover, it provides an explicit construction of the associated field. Concrete field realizations of common flow-based approaches, such as stochastic interpolants, diffusion models and flow matching, are summarized in Appendix A, Table 3.

### 3.3. From Fields to Flows

We now establish the converse field-to-flow direction of the duality. Motivated by Theorem 3.1, a natural attempt to recover a Conditional Flow from a given forward-only Interaction Field $\mathbf{E}^{\mathbf{x}_0,\mathbf{x}_T}(\mathbf{x}, t)$ is to set

$$\mathbf{v}_t^{\mathbf{x}_0,\mathbf{x}_T}(\mathbf{x}) = \frac{\mathbf{E}^{\mathbf{x}_0,\mathbf{x}_T}(\mathbf{x},t)_{\mathbf{x}}}{\mathbf{E}^{\mathbf{x}_0,\mathbf{x}_T}(\mathbf{x},t)_t}, \quad p_t^{\mathbf{x}_0,\mathbf{x}_T}(\mathbf{x}) = \mathbf{E}^{\mathbf{x}_0,\mathbf{x}_T}(\mathbf{x},t)_t. \quad (25)$$

However, the last component $\mathbf{E}^{\mathbf{x}_0,\mathbf{x}_T}(\mathbf{x}, t)_t$ of a forward-only Interaction Field is not required to be normalized:

$$\Phi_t^{\mathbf{x}_0,\mathbf{x}_T} := \int \mathbf{E}^{\mathbf{x}_0,\mathbf{x}_T}(\mathbf{x}, t)_t \, d\mathbf{x} \neq 1. \quad (26)$$

Here, $\Phi_t^{\mathbf{x}_0,\mathbf{x}_T}$ is a scalar that may, a priori, depend on $\mathbf{x}_0, \mathbf{x}_T$ and $t$. Nevertheless, it is in fact independent of both, as formalized in our following proposition.

**Proposition 3.2.** (Flux Conservation across Slices). *The scalar $\Phi_t^{\mathbf{x}_0,\mathbf{x}_T}$ in (26) is the flux through the slice $\{t = \text{const}\}$ at distance $t$ from the left plate and coincides with the total flux $\Phi_0$ (8) of the forward-only Interaction Field.*

Having established this proposition, we now formalize the field-to-flow direction of the duality.

**Theorem 3.3.** (forward-only IFM $\subseteq$ CFM). *Let $\pi_0$ and $\pi_T$ be D-dimensional target and source distributions, respectively. Let $\mathbf{E}^{\mathbf{x}_0,\mathbf{x}_T}(\mathbf{x},t)$ be a forward-only Interaction Field, and let $\mathbf{E}(\mathbf{x},t)$ denote the global field obtained by via generalized superposition principle (9). Define the conditional velocity and conditional density path by*

$$\mathbf{v}_t^{\mathbf{x}_0,\mathbf{x}_T}(\mathbf{x})=\frac{\mathbf{E}^{\mathbf{x}_0,\mathbf{x}_T}(\mathbf{x},t)_\mathbf{x}}{\mathbf{E}^{\mathbf{x}_0,\mathbf{x}_T}(\mathbf{x},t)_t}, \; p_t^{\mathbf{x}_0,\mathbf{x}_T}(\mathbf{x})=\frac{\mathbf{E}^{\mathbf{x}_0,\mathbf{x}_T}(\mathbf{x},t)_t}{\Phi_0}. \quad (27)$$

*Then, under mild assumptions specified in Appendix B.3:*

1. *The pair $(\mathbf{v}_t^{\mathbf{x}_0,\mathbf{x}_T}, p_t^{\mathbf{x}_0,\mathbf{x}_T})$ is a Conditional Flow (§2.1).*

2. *The global drift $\mathbf{v}_t$ (4) and probability path $p_t$ (3) can be recovered from the global field $\mathbf{E}(\mathbf{x},t)$ via*

$$\mathbf{v}_t(\mathbf{x}) = \frac{\mathbf{E}(\mathbf{x},t)_\mathbf{x}}{\mathbf{E}(\mathbf{x},t)_t}, \quad p_t(\mathbf{x}) = \frac{\mathbf{E}(\mathbf{x},t)_t}{\Phi_0}. \quad (28)$$

3. *The CFM generative dynamics (4) coincide with the dynamics induced by field lines (10), reparameterized by the distance from the plate (12):*

$$\frac{d\mathbf{x}}{dt} = \mathbf{v}_t(\mathbf{x}) = \frac{\mathbf{E}(\mathbf{x},t)_\mathbf{x}}{\mathbf{E}(\mathbf{x},t)_t}. \quad (29)$$

Our Theorem 3.3 show that any any forward-only Interaction Field can be represented as a Conditional Flow inducing the same generative dynamics. Moreover, the theorem provides an explicit construction of the corresponding flow.

**Example (PFGM's flow-based construction).** Previously, we showed that PFGM is a forward-only one-sided IFM. Therefore, by Theorem 3.3, it admits an equivalent Conditional Flow representation. This construction matches the objects introduced in §2.4 as candidates for a flow-based interpretation of PFGM. Consequently, at least in principle, the theoretical results derived in PFGM++ can be recovered by applying Theorem 3.3 to the special case of PFGM.

**Example (IFM's flow-based construction).** In §2.2, the canonical IFM field is built as a forward-only Interaction Field via a rather involved, bespoke construction. Using our Theorem 3.3, we can express it as a Conditional Flow, which has clearer form: $\mathbf{v}_t^{\mathbf{x}_0,\mathbf{x}_T}(\mathbf{x}_t) = \dot{\mathcal{I}}(t,\mathbf{x}_0,\mathbf{x}_T) + \frac{\dot{\sigma}(t)}{\sigma(t)}(\mathbf{x}_t - \mathcal{I}(t,\mathbf{x}_0,\mathbf{x}_T))$, $p_t^{\mathbf{x}_0,\mathbf{x}_T}(\mathbf{x}) = \mathcal{N}(\mathbf{x} \mid \mathcal{I}(t,\mathbf{x}_0,\mathbf{x}_T), \sigma(t)^2 I)$ where $\mathcal{I}(t,\mathbf{x}_0,\mathbf{x}_T) = (1 - t/L)\,\mathbf{x}_0 + (t/L)\,\mathbf{x}_T$. This representation makes explicit that the IFM field is induced by the two-sided stochastic interpolant from §2.1, namely, $\mathbf{x}_t = \mathcal{I}(t,\mathbf{x}_0,\mathbf{x}_T) + \sigma(t)\epsilon$. Additional field–flow construction examples are summarized in Appendix A, Table 3.

### 3.4. Duality Takeaways

In this section, we discuss theoretical insights implied by the established duality. First, the duality implies that CFM and forward-only IFM induce identical generative dynamics and provides explicit mappings between the two; see (22) and (27). This naturally raises the question: are there IFM dynamics that cannot be represented within CFM? The answer is **yes**. Indeed, allowing backward-oriented field lines (as in EFM) goes beyond the forward-only setting.

> **Takeaway 1 (CFM and IFM expressiveness):**
>
> $$\text{CFM} \equiv \text{forward-only IFM} \subsetneq \text{IFM}$$

Second, viewing forward-only IFM through the lens of the CFM in (28) yields a natural probabilistic interpretation.

> **Takeaway 2 (IFM probabilistic interpretation).** The generative dynamics (12) defined by a global forward-only Interaction Field $\mathbf{E}(\mathbf{x},t)$ induce the distribution path $p_t(\mathbf{x}) = \mathbf{E}(\mathbf{x},t)_t/\Phi_0$.

Third, the CFM-based construction (22) suggests a CFM-style objective for learning the forward-only data-space dynamics (12). In particular, the drift $\mathbf{v}_t := \mathbf{E}(\mathbf{x},t)_\mathbf{x}/\mathbf{E}(\mathbf{x},t)_t$ can be obtained as a minimizer of

$$\mathbb{E}_{t,\mathbf{x}_0,\mathbf{x}_T,\mathbf{x}} \left\| \mathbf{v}_t(\mathbf{x}) - \frac{\mathbf{E}^{\mathbf{x}_0,\mathbf{x}_T}(\mathbf{x},t)_\mathbf{x}}{\mathbf{E}^{\mathbf{x}_0,\mathbf{x}_T}(\mathbf{x},t)_t} \right\|_2^2, \quad (30)$$

where $\mathbf{x} \sim p_t^{\mathbf{x}_0,\mathbf{x}_T}(\mathbf{x}) = \mathbf{E}^{\mathbf{x}_0,\mathbf{x}_T}(\mathbf{x},t)_t/\Phi_0$.

> **Takeaway 3 (IFM field-informed volume coverage).** Forward-only IFM induces a natural volume-coverage distribution over data space, encoded by its last component: $p_t^{\mathbf{x}_0,\mathbf{x}_T}(\mathbf{x}) = \mathbf{E}^{\mathbf{x}_0,\mathbf{x}_T}(\mathbf{x},t)_t/\Phi_0$.

## 4. Practical Takeaways and Discussion

So far, we have discussed two theoretical implications of our duality: **(a)** it yields a probabilistic interpretation of the forward-only IFM dynamics, and **(b)** it shows that forward-only IFM coincides with CFM and can be developed within a unified framework. The latter point is particularly timely given the recent surge of interest in physics-inspired generative models (Liu et al., 2023), including electrostatics and field theory, as in our setting. Beyond interpretation, a more practical question is: *What can the IFM viewpoint improve in conventional CFM training and inference?*

**Multi-sample velocity estimation formula.** The IFM framework naturally suggests the following CFM velocity

estimator using global field–velocity relation in (28):

$$\hat{\mathbf{v}}_t^N(\mathbf{x}) := \frac{\hat{\mathbf{E}}^N(\mathbf{x},t)_{\mathbf{x}}}{\hat{\mathbf{E}}^N(\mathbf{x},t)_t} = \frac{1/N \sum_{i=1}^N \mathbf{E}^{\mathbf{x}_0^i,\mathbf{x}_T^i}(\mathbf{x})_{\mathbf{x}}}{1/N \sum_{i=1}^N \mathbf{E}^{\mathbf{x}_0^i,\mathbf{x}_T^i}(\mathbf{x})_t} =$$

$$= \frac{\sum_{i=1}^N \mathbf{v}_t^{\mathbf{x}_0^i,\mathbf{x}_T^i}(\mathbf{x}) p_t^{\mathbf{x}_0^i,\mathbf{x}_T^i}(\mathbf{x})}{\sum_{i=1}^N p_t^{\mathbf{x}_0^i,\mathbf{x}_T^i}(\mathbf{x})} = \sum_{i=1}^N \omega_i \mathbf{v}_t^{\mathbf{x}_0^i,\mathbf{x}_T^i}(\mathbf{x}), \quad (31)$$

where $\omega_i := p_t^{\mathbf{x}_0^i,\mathbf{x}_T^i}(\mathbf{x}) \big/ \sum_{k=1}^N p_t^{\mathbf{x}_0^k,\mathbf{x}_T^k}(\mathbf{x})$, $\mathbf{x}_0^i, \mathbf{x}_T^i \sim \pi_{0,T}$, and $N$ is the number of samples used for velocity estimation. This estimator can be substituted into any expression that uses the global velocity, without requiring additional learning. For example, it may benefit the recently proposed Mean Flow models (Geng et al., 2025), whose objective depends on the global velocity; in practice, however, the authors use a single-sample estimator, which can lead to slower convergence and worse final performance. To begin understanding the limits of this multi-sample estimator, we use it as the target in the CFM objective, replacing the standard single-sample target:

$$\mathbb{E}_{t,\mathbf{x}_0,\mathbf{x}_T,\mathbf{x}_t}\left\|\mathbf{v}_t(\mathbf{x}_t) - \sum_{i=1}^N \omega_i \mathbf{v}_t^{\mathbf{x}_0^i,\mathbf{x}_T^i}(\mathbf{x}_t)\right\|_2^2, \quad (32)$$

where $\mathbf{x}_0^1, \mathbf{x}_T^1 = \mathbf{x}_0, \mathbf{x}_T$ is used to obtain the intermediate point $\mathbf{x}_t \sim p_t^{\mathbf{x}_0,\mathbf{x}_T}$. When $N = 1$, this target reduces to the original single-sample CFM objective. We note that recent works (Bertrand et al., 2025; Ryzhakov et al., 2024) consider multi-sample targets for the specific Flow Matching framework, whereas our duality naturally suggests a multi-sample estimator for the broader CFM framework.

*Multi-sample target in practice.* In our experiments, we consider the two-sided linear interpolant from (Albergo et al., 2023), EDM (Karras et al., 2022), and PFGM++ (Xu et al., 2023), with auxiliary-dimension hyperparameter $d \in \{128, 2048\}$, on the CIFAR-10 unconditional generation task; for additional details, see Appendix C. Table 2 provides qualitative results for different numbers of samples used in the multi-sample estimate. Overall, the multi-sample target yields a slight benefit for EDM and PFGM++, but provides no improvement for the considered two-sided interpolant.

*Table 2.* Interpolant quality (measured by FID (Heusel et al., 2018)) as a function of the number of samples $N$ used for target estimation. For each model (column), the best score is **bolded**.

| $N$ | Two-sided Linear | EDM | PFGM++ (d=128) | PFGM++ (d=2048) |
|---|---|---|---|---|
| 1 | **3.03** | 2.29 | 2.40 | 2.40 |
| 256 | 3.04 | 2.21 | **2.24** | **2.14** |
| 2048 | 3.05 | **2.12** | 2.28 | 2.19 |

*Dominant sample.* To better understand this behavior, we examine the multi-sample estimator in (31) more closely. We find that the weight distribution $\{\omega_i\}_{i=1}^N$ in (31) is highly concentrated: a single conditional velocity $\mathbf{v}_t^{\mathbf{x}_0^i,\mathbf{x}_T^i}(\mathbf{x}_t)$ typically dominates the weighted sum. In particular, the dom-

inant term is always $i = 1$, i.e., the same conditional pair $\mathbf{x}_0^1, \mathbf{x}_T^1 = \mathbf{x}_0, \mathbf{x}_T$ that was used to generate the intermediate point $\mathbf{x}_t \sim p_t^{\mathbf{x}_0,\mathbf{x}_T}$ at which the global velocity is estimated. To test the importance of this *dominant* pair, we remove it from the target estimator in (31); this leads to a dramatic degradation in performance (FID $> 100$).

*Weight distribution analysis.* Motivated by these observations, we quantify the concentration of the weights $\{\omega_i\}_{i=1}^N$ using the Gini coefficient, which is zero for a degenerate distribution and attains its maximum for a uniform distribution. Figure 2 shows that the Gini coefficient increases with the number of samples $N$ for all one-sided interpolants, but remains at (or near) zero for our two-sided linear interpolant. This trend mirrors the FID results in Table 2. Specifically, for one-sided interpolants, both the FID and the Gini coefficient increase as $N$ grows, whereas for the two-sided interpolant, both metrics remain essentially unchanged across different values of $N$.

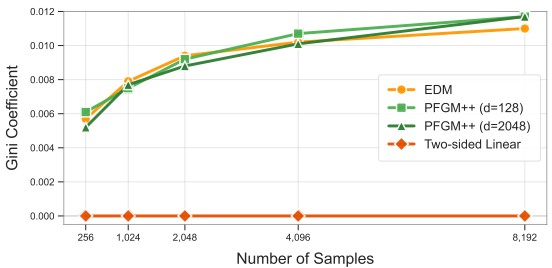

*Figure 2.* Dependence of the Gini coefficient of $\{\omega_i\}_{i=1}^N$ on the number of samples $N$ for different interpolants.

An additional implication of the increasing Gini coefficient is that the weight distribution becomes less degenerate as $N$ grows. One interpretation is that, with more samples, the estimator is more likely to include conditional $\mathbf{x}_0, \mathbf{x}_T$ that make a non-negligible contribution to the final estimate. This suggests a *promising research direction for reducing the variance of multi-sample estimators*: instead of sampling conditional pairs uniformly from the dataset, one could preferentially select pairs that are more likely to be relevant at $\mathbf{x}$, thereby producing a less concentrated set of weights.

**Volume coverage distribution.** Another practical implication suggested by the IFM viewpoint is to use an alternative volume-coverage distribution $p_{\text{vol}}(\mathbf{x}, t)$ in place of, or in addition to, the default CFM distribution $p_t(\mathbf{x}) \times \mathcal{U}[0,T](t)$ induced by the path distribution. At present, it is unclear which choice of $p_{\text{vol}}(\mathbf{x}, t)$ is preferable; answering this question requires further study. The optimal distribution is likely task-dependent, and may be useful when it increases coverage of regions where the model exhibits systematic errors. However, such a customized volume-coverage distribution is only viable once we can construct a reliable multi-sample velocity estimator; otherwise, the resulting objective may become strongly biased. *This further underscores the importance of accurate multi-sample velocity estimation.*

## Impact Statements

This paper presents work whose goal is to advance the field of machine learning. There are many potential societal consequences of our work, none of which we feel must be specifically highlighted here.

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

# A. CFM and IFM construction examples

## A.1. Flow-field Dual Constructions

In this section, we provide additional examples of common frameworks formulated via the flow-field dual construction, as summarized in Table 3.

*Table 3.* **Flow-field dual constructions**. This table shows how a common framework can be formulated using both CFM and IFM.

| Framework | CFM structure $\left(\mathbf{v}_t^{\mathbf{x}_0,\mathbf{x}_T}(\mathbf{x}),\, p_t^{\mathbf{x}_0,\mathbf{x}_T}(\mathbf{x})\right)$ | IFM structure $\mathbf{E}^{\mathbf{x}_0,\mathbf{x}_T}(\tilde{\mathbf{x}}) = (\mathbf{E}^{\mathbf{x}_0,\mathbf{x}_T}(\tilde{\mathbf{x}})_{\mathbf{x}}, \mathbf{E}^{\mathbf{x}_0,\mathbf{x}_T}(\tilde{\mathbf{x}})_t),\, \tilde{\mathbf{x}}=(\mathbf{x},t)$ |
|---|---|---|
| **Flow Matching** (Lipman et al., 2022) | $\mathbf{v}_t^{\mathbf{x}_0}(\mathbf{x}) = \dfrac{\mathbf{x}-\mathbf{x}_0}{t}$ 
 $p_t^{\mathbf{x}_0}(\mathbf{x}) = \mathcal{N}\left(\mathbf{x} \mid (1-t)\mathbf{x}_0,\, t^2 I\right)$ | $\mathbf{E}^{\mathbf{x}_0}(\mathbf{x},t)_{\mathbf{x}} = \mathbf{v}_t^{\mathbf{x}_0}(\mathbf{x})\, p_t^{\mathbf{x}_0}(\mathbf{x})$ 
 $\mathbf{E}^{\mathbf{x}_0}(\mathbf{x},t)_t = p_t^{\mathbf{x}_0}(\mathbf{x})$ |
| **VE-diffusion** (Song et al., 2020) | $\mathbf{v}_t^{\mathbf{x}_0}(\mathbf{x}) = \mathbf{x}-\mathbf{x}_0$ 
 $p_t^{\mathbf{x}_0}(\mathbf{x}) = \mathcal{N}\left(\mathbf{x} \mid (1-t)\mathbf{x}_0,\, t^2 I\right)$ | $\mathbf{E}^{\mathbf{x}_0}(\mathbf{x},t)_{\mathbf{x}} = \mathbf{v}_t^{\mathbf{x}_0}(\mathbf{x})\, p_t^{\mathbf{x}_0}(\mathbf{x})$ 
 $\mathbf{E}^{\mathbf{x}_0}(\mathbf{x},t)_t = p_t^{\mathbf{x}_0}(\mathbf{x})$ |
| **Linear Interpolant** (Albergo et al., 2023) | $\mathbf{v}_t^{\mathbf{x}_0,\mathbf{x}_T}(\mathbf{x}) = \dfrac{\dot{\sigma}(t)}{\sigma(t)}(\mathbf{x}-\mathcal{I}(t,\mathbf{x}_0,\mathbf{x}_T))+\dot{\mathcal{I}}(t,\mathbf{x}_0,\mathbf{x}_T)$ 
 $p_t^{\mathbf{x}_0,\mathbf{x}_T}(\mathbf{x}) = \mathcal{N}(x \mid \mathcal{I}(t,\mathbf{x}_0,\mathbf{x}_T),\, \sigma(t)^2 I)$ 

 $\mathcal{I}(t,\mathbf{x}_0,\mathbf{x}_T) = \mathbf{x}_0 \cdot (1-t/T) + \mathbf{x}_T \cdot t/T$ 
 $\sigma(t) = \sqrt{2 \cdot t/T \cdot (1-t/T)}$ | $\mathbf{E}^{\mathbf{x}_0,\mathbf{x}_T}(\mathbf{x},t)_{\mathbf{x}} = \mathbf{v}_t^{\mathbf{x}_0,\mathbf{x}_T}(\mathbf{x})\, p_t^{\mathbf{x}_0,\mathbf{x}_T}(\mathbf{x})$ 
 $\mathbf{E}^{\mathbf{x}_0,\mathbf{x}_T}(\mathbf{x},t)_t = p_t^{\mathbf{x}_0,\mathbf{x}_T}(\mathbf{x})$ |
| **PFGM** (Xu et al., 2022a) | $\mathbf{v}_t^{\mathbf{x}_0}(\mathbf{x}) = \dfrac{\mathbf{x}-\mathbf{x}_0}{t}$ 
 $p_t^{\mathbf{x}_0}(\mathbf{x}) \propto t/\left(\|\mathbf{x}-\mathbf{x}_0\| + t^2\right)^{\frac{D+1}{2}}$ | $\mathbf{E}^{\mathbf{x}_0}(\tilde{\mathbf{x}}) = \dfrac{\tilde{\mathbf{x}}-\tilde{\mathbf{x}}_0}{\|\tilde{\mathbf{x}}-\tilde{\mathbf{x}}_0\|_2^{D+1}}$ |
| **PFGM++** ($d \geq 1$) (Xu et al., 2023) | $\mathbf{v}_t^{\mathbf{x}_0}(\mathbf{x}) = \dfrac{\mathbf{x}-\mathbf{x}_0}{t}$ 
 $p_t^{\mathbf{x}_0}(\mathbf{x}) \propto t/\left(\|\mathbf{x}-\mathbf{x}_0\| + t^2\right)^{\frac{D+d}{2}}$ | $\mathbf{E}^{\mathbf{x}_0}(\mathbf{x},t)_{\mathbf{x}} = \mathbf{v}_t^{\mathbf{x}_0}(\mathbf{x})\, p_t^{\mathbf{x}_0}(\mathbf{x})$ 
 $\mathbf{E}^{\mathbf{x}_0}(\mathbf{x},t)_t = p_t^{\mathbf{x}_0}(\mathbf{x})$ |
| **IFM** (Manukhov et al., 2026) | $\mathbf{v}_t^{\mathbf{x}_0,\mathbf{x}_T}(\mathbf{x}) = \dfrac{\dot{\sigma}(t)}{\sigma(t)}(\mathbf{x}-\mathcal{I}(t,\mathbf{x}_0,\mathbf{x}_T))+\dot{\mathcal{I}}(t,\mathbf{x}_0,\mathbf{x}_T)$ 
 $p_t^{\mathbf{x}_0,\mathbf{x}_T}(\mathbf{x}) = \mathcal{N}(x \mid \mathcal{I}(t,\mathbf{x}_0,\mathbf{x}_T),\, \sigma(t)^2 I)$ 

 $\mathcal{I}(t,\mathbf{x}_0,\mathbf{x}_T) = \mathbf{x}_0 \cdot (1-t/T) + \mathbf{x}_T \cdot t/T$ 
 $\sigma(t) = \sin\left(2\pi t/T\right)$ | $\mathbf{E}^{\mathbf{x}_0,\mathbf{x}_T}(\mathbf{x},t)_{\mathbf{x}} = \mathbf{v}_t^{\mathbf{x}_0,\mathbf{x}_T}(\mathbf{x})\, p_t^{\mathbf{x}_0,\mathbf{x}_T}(\mathbf{x})$ 
 $\mathbf{E}^{\mathbf{x}_0,\mathbf{x}_T}(\mathbf{x},t)_t = p_t^{\mathbf{x}_0,\mathbf{x}_T}(\mathbf{x})$ 

 OR originally defined as in §2.2 |
| **EFM** (Kolesov et al., 2025) | $\times$ | $\mathbf{E}^{\mathbf{x}_0,\mathbf{x}_T}(\tilde{\mathbf{x}}) = \dfrac{\tilde{\mathbf{x}}-\tilde{\mathbf{x}}_0}{\|\tilde{\mathbf{x}}-\tilde{\mathbf{x}}_0\|_2^{D+1}} - \dfrac{\tilde{\mathbf{x}}-\tilde{\mathbf{x}}_T}{\|\tilde{\mathbf{x}}-\tilde{\mathbf{x}}_T\|_2^{D+1}}$ |

## A.2. Original IFM field realization

For completeness, we restate the forward-only Interaction Field realization proposed in Manukhov et al. (2026, Appendix A.4). The construction specifies the magnitude $\|\mathbf{E}^{\mathbf{x}_0,\mathbf{x}_L}(\tilde{\mathbf{x}})\|_2$ and the unit direction $\mathbf{n}^{\mathbf{x}_0,\mathbf{x}_L}(\tilde{\mathbf{x}})$ of the conditional field $\mathbf{E}^{\mathbf{x}_0,\mathbf{x}_L}(\tilde{\mathbf{x}})$, where $\tilde{\mathbf{x}} = (\mathbf{x}, z)$. Concretely, they define

$$\|\mathbf{E}^{\mathbf{x}_0,\mathbf{x}_L}(\tilde{\mathbf{x}})\|_2 := \frac{\exp\left(-\frac{r_\perp(\tilde{\mathbf{x}})^2}{2\sigma(z)^2}\right)}{\sigma(z)^D \cos\alpha(\tilde{\mathbf{x}})}, \tag{33}$$

$$\mathbf{n}^{\mathbf{x}_0,\mathbf{x}_L}(\tilde{\mathbf{x}}) := \mathbf{e}_\perp \cdot \sin\alpha(\tilde{\mathbf{x}}) + \mathbf{e}'_z \cdot \cos\alpha(\tilde{\mathbf{x}}), \tag{34}$$

$$\mathbf{e}_\perp := \frac{\mathbf{x}_\perp(\tilde{\mathbf{x}})}{r_\perp(\tilde{\mathbf{x}})}, \qquad \mathbf{e}'_z := \frac{\tilde{\mathbf{x}}_L - \tilde{\mathbf{x}}_0}{\|\tilde{\mathbf{x}}_L - \tilde{\mathbf{x}}_0\|_2}, \tag{35}$$

where $\tilde{\mathbf{x}}_0 = (\mathbf{x}_0, 0)$ and $\tilde{\mathbf{x}}_L = (\mathbf{x}_L, L)$ denote the endpoints, and

$$\sigma(z) := \sin(2\pi z/L)$$

(for $d = L/2$ in their notation). The transverse component is defined as

$$\mathbf{x}_\perp(\tilde{\mathbf{x}}) := \mathbf{x} - \mathbf{x}_0(1 - z/L) - \mathbf{x}_L(z/L), \qquad r_\perp(\tilde{\mathbf{x}}) := \|\mathbf{x}_\perp(\tilde{\mathbf{x}})\|_2.$$

Finally, the angle $\alpha(\tilde{\mathbf{x}})$ is chosen so that the ratio $r_\perp(\tilde{\mathbf{x}})/\sigma(z)$ remains constant along each field line, which yields

$$\alpha(\tilde{\mathbf{x}}) := \arctan\left(\frac{2\pi}{L} r_\perp(\mathbf{x}, z) \cot\left(\frac{2\pi}{L} z\right)\right).$$

# B. Proofs

In this section we provide proofs for all theorems and propositions from the main part of the paper.

## B.1. Assumptions for Theorem 3.1

For all $t \in (0, T)$, $\mathbf{x} \in \mathbb{R}^D$:

1. $p_t^{\mathbf{x}_0, \mathbf{x}_T}(\mathbf{x}) \in C^1((0, T) \times \mathbb{R}^D)$ is strictly positive;

2. All integrals appearing in the proof (those involving $p_t^{\mathbf{x}_0, \mathbf{x}_T}$ and $\mathbf{v}_t^{\mathbf{x}_0, \mathbf{x}_T} p_t^{\mathbf{x}_0, \mathbf{x}_T}$) converge;

3. $\mathbf{v}_t^{\mathbf{x}_0, \mathbf{x}_T}(\mathbf{x})$ is locally Lipschitz continuous in $\mathbf{x}$, uniformly in $t$ on compact subsets of $(0, T) \times \mathbb{R}^D$.

## B.2. Proof of Theorem 3.1

We prove the three items in order.

1. **$\mathbf{E}^{\mathbf{x}_0, \mathbf{x}_T}(\mathbf{x}, t)$ is a forward-only Interaction Field with $\Phi_0 = 1$.**

   *Forward-only.* By definition of the field,

   $$\mathbf{E}^{\mathbf{x}_0, \mathbf{x}_T}(\mathbf{x}, t)_t = p_t^{\mathbf{x}_0, \mathbf{x}_T}(\mathbf{x}) \geq 0.$$

   Moreover, if $\mathbf{E}^{\mathbf{x}_0, \mathbf{x}_T}(\mathbf{x}, t) \neq \mathbf{0}$, then in particular $\mathbf{E}^{\mathbf{x}_0, \mathbf{x}_T}(\mathbf{x}, t)_t \neq 0$, hence $\mathbf{E}^{\mathbf{x}_0, \mathbf{x}_T}(\mathbf{x}, t)_t > 0$ at every point where the field is nonzero. Therefore $\mathbf{E}^{\mathbf{x}_0, \mathbf{x}_T}$ is forward-only field.

   *Field lines connect the two particles.* By the superposition principle (Bogachev et al., 2021) for the continuity equation there are exist absolutely continuous curves $\mathbf{x}(t)$ satisfying

   $$\frac{d\mathbf{x}}{dt} = \mathbf{v}_t^{\mathbf{x}_0, \mathbf{x}_T}(\mathbf{x}), \quad \mathbf{x}(0) = \mathbf{x}_0, \quad \mathbf{x}(T) = \mathbf{x}_T.$$

   Under the Lipschitz assumption, this ODE with initial conditions has a *unique* solution. Therefore the support of $\mathbf{x}(t)$ consists of a single curve. So the curve $\tilde{\mathbf{x}}(t) := (\mathbf{x}(t), t)$ connects $\tilde{\mathbf{x}}_0 = (\mathbf{x}_0, 0)$ to $\tilde{\mathbf{x}}_T = (\mathbf{x}_T, T)$ in $\mathbb{R}^{D+1}$ and follows

   $$\frac{d\tilde{\mathbf{x}}}{dt} = (\mathbf{v}_t^{\mathbf{x}_0, \mathbf{x}_T}(\mathbf{x}(t)), 1).$$

   Now note that

   $$\mathbf{E}^{\mathbf{x}_0, \mathbf{x}_T}(\mathbf{x}, t) = \left(\mathbf{v}_t^{\mathbf{x}_0, \mathbf{x}_T}(\mathbf{x}) \, p_t^{\mathbf{x}_0, \mathbf{x}_T}(\mathbf{x}), \; p_t^{\mathbf{x}_0, \mathbf{x}_T}(\mathbf{x})\right) = p_t^{\mathbf{x}_0, \mathbf{x}_T}(\mathbf{x}) \, (\mathbf{v}_t^{\mathbf{x}_0, \mathbf{x}_T}(\mathbf{x}), 1).$$

   Hence $\mathbf{E}^{\mathbf{x}_0, \mathbf{x}_T}(\tilde{\mathbf{x}})$ is a positive scalar multiple of $(\mathbf{v}_t^{\mathbf{x}_0, \mathbf{x}_T}(\mathbf{x}), 1)$ on $p_t^{\mathbf{x}_0, \mathbf{x}_T}(\mathbf{x}) > 0$, so it has the same integral curves up to a reparameterization. Concretely, define a new parameter $\tau$ along the above curve by

   $$\frac{dt}{d\tau} = p_t^{\mathbf{x}_0, \mathbf{x}_T}(\mathbf{x}(t)) \qquad \left(\text{equivalently } \frac{d\tau}{dt} = \frac{1}{p_t^{\mathbf{x}_0, \mathbf{x}_T}(\mathbf{x}(t))}\right),$$

   and set $\tilde{\mathbf{x}}(\tau) := \tilde{\mathbf{x}}(t(\tau))$. Then

   $$\frac{d\tilde{\mathbf{x}}}{d\tau} = \frac{d\tilde{\mathbf{x}}}{dt}\frac{dt}{d\tau} = (\mathbf{v}_t^{\mathbf{x}_0, \mathbf{x}_T}(\mathbf{x}(t)), 1) \, p_t^{\mathbf{x}_0, \mathbf{x}_T}(\mathbf{x}(t)) = \left(\mathbf{v}_t^{\mathbf{x}_0, \mathbf{x}_T}(\mathbf{x}(t)) p_t^{\mathbf{x}_0, \mathbf{x}_T}(\mathbf{x}(t)), \; p_t^{\mathbf{x}_0, \mathbf{x}_T}(\mathbf{x}(t))\right) = \mathbf{E}^{\mathbf{x}_0, \mathbf{x}_T}(\tilde{\mathbf{x}}(\tau)).$$

   Therefore $\tilde{\mathbf{x}}(\tau)$ is a field line of $\mathbf{E}^{\mathbf{x}_0, \mathbf{x}_T}$ connecting $\tilde{\mathbf{x}}_0$ to $\tilde{\mathbf{x}}_T$.

   *Divergence-free.* Using $\mathbf{E}^{\mathbf{x}_0, \mathbf{x}_T}(\mathbf{x}, t)_{\mathbf{x}} = \mathbf{v}_t^{\mathbf{x}_0, \mathbf{x}_T}(\mathbf{x}) \, p_t^{\mathbf{x}_0, \mathbf{x}_T}(\mathbf{x})$ and $p_t^{\mathbf{x}_0, \mathbf{x}_T}(\mathbf{x}) = \mathbf{E}^{\mathbf{x}_0, \mathbf{x}_T}(\mathbf{x}, t)_t$, we obtain

   $$\nabla_{(\mathbf{x}, t)} \cdot \mathbf{E}^{\mathbf{x}_0, \mathbf{x}_T}(\mathbf{x}, t) = \partial_t \mathbf{E}^{\mathbf{x}_0, \mathbf{x}_T}(\mathbf{x}, t)_t + \nabla_{\mathbf{x}} \cdot \mathbf{E}^{\mathbf{x}_0, \mathbf{x}_T}(\mathbf{x}, t)_{\mathbf{x}}$$
   $$= \partial_t p_t^{\mathbf{x}_0, \mathbf{x}_T}(\mathbf{x}) + \nabla_{\mathbf{x}} \cdot \left(\mathbf{v}_t^{\mathbf{x}_0, \mathbf{x}_T}(\mathbf{x}) \, p_t^{\mathbf{x}_0, \mathbf{x}_T}(\mathbf{x})\right) = 0,$$

where the last equality is the continuity equation for the Conditional Flow.

*Fixed total flux and $\Phi_0 = 1$.* Let $\partial M$ be any closed surface that encloses $\tilde{\mathbf{x}}_0 = (\mathbf{x}_0, 0)$ but not $\tilde{\mathbf{x}}_T = (\mathbf{x}_T, T)$. We show that $\int_{\partial M} \mathbf{E}^{\mathbf{x}_0, \mathbf{x}_T} \cdot d\mathbf{S} = 1$, which identifies the total flux as $\Phi_0 = 1$ and shows it is independent of $(\mathbf{x}_0, \mathbf{x}_T)$.

First, let's reduce the flux on $\partial M$ to the flux on a small pillbox around $\tilde{\mathbf{x}}_0$. Since $\tilde{\mathbf{x}}_0$ lies in the interior of $M$, we can choose $R > 0$ and $\epsilon \in (0, T)$ such that the pillbox

$$P := B_R(\mathbf{x}_0) \times [-\epsilon, \epsilon]$$

is contained in $M$ (and in particular does not contain $\tilde{\mathbf{x}}_T$ because $\epsilon < T$). Consider the region $M \setminus P$. It contains neither endpoint, so $\mathbf{E}^{\mathbf{x}_0, \mathbf{x}_T}$ is divergence-free there. Hence, by the divergence theorem,

$$0 = \int_{M \setminus P} \nabla \cdot \mathbf{E}^{\mathbf{x}_0, \mathbf{x}_T} = \int_{\partial(M \setminus P)} \mathbf{E}^{\mathbf{x}_0, \mathbf{x}_T} \cdot d\mathbf{S} = \int_{\partial M} \mathbf{E}^{\mathbf{x}_0, \mathbf{x}_T} \cdot d\mathbf{S} - \int_{\partial P} \mathbf{E}^{\mathbf{x}_0, \mathbf{x}_T} \cdot d\mathbf{S},$$

where the minus sign comes from the fact that the normal on $\partial P$ is inward for the region $M \setminus P$. Therefore,

$$\int_{\partial M} \mathbf{E}^{\mathbf{x}_0, \mathbf{x}_T} \cdot d\mathbf{S} = \int_{\partial P} \mathbf{E}^{\mathbf{x}_0, \mathbf{x}_T} \cdot d\mathbf{S}.$$

Next, let's compute the flux through the pillbox $\partial P$. Decompose $\partial P$ into bottom, lateral, and top parts:

$$\partial P = \underbrace{B_R(\mathbf{x}_0) \times \{-\epsilon\}}_{\text{bottom}} \cup \underbrace{\partial B_R(\mathbf{x}_0) \times [-\epsilon, \epsilon]}_{\text{lateral}} \cup \underbrace{B_R(\mathbf{x}_0) \times \{\epsilon\}}_{\text{top}}.$$

*Bottom.* By the forward-only convention, $\mathbf{E}^{\mathbf{x}_0, \mathbf{x}_T}(\mathbf{x}, t) = 0$ for $t < 0$, hence

$$\int_{B_R \times \{-\epsilon\}} \mathbf{E}^{\mathbf{x}_0, \mathbf{x}_T} \cdot d\mathbf{S} = -\int_{B_R} \mathbf{E}^{\mathbf{x}_0, \mathbf{x}_T}(\mathbf{x}, -\epsilon)_t \, d\mathbf{x} = 0.$$

*Lateral.* For $t \in (0, \epsilon)$ we have $\mathbf{E}^{\mathbf{x}_0, \mathbf{x}_T}(\mathbf{x}, t)_{\mathbf{x}} = \mathbf{v}_t^{\mathbf{x}_0, \mathbf{x}_T}(\mathbf{x}) \, p_t^{\mathbf{x}_0, \mathbf{x}_T}(\mathbf{x})$ and $\mathbf{E}^{\mathbf{x}_0, \mathbf{x}_T}(\mathbf{x}, t)_t = p_t^{\mathbf{x}_0, \mathbf{x}_T}(\mathbf{x})$. Integrate the continuity equation $\partial_t p_t^{\mathbf{x}_0, \mathbf{x}_T} + \nabla_{\mathbf{x}} \cdot (\mathbf{v}_t^{\mathbf{x}_0, \mathbf{x}_T} p_t^{\mathbf{x}_0, \mathbf{x}_T}) = 0$ over $B_R(\mathbf{x}_0)$ and apply the divergence theorem in $\mathbf{x}$:

$$\frac{d}{dt} \int_{B_R(\mathbf{x}_0)} p_t^{\mathbf{x}_0, \mathbf{x}_T}(\mathbf{x}) \, d\mathbf{x} = -\int_{\partial B_R(\mathbf{x}_0)} \mathbf{v}_t^{\mathbf{x}_0, \mathbf{x}_T}(\mathbf{x}) \, p_t^{\mathbf{x}_0, \mathbf{x}_T}(\mathbf{x}) \cdot \mathbf{n} \, dS.$$

Integrating from $t = 0$ to $t = \epsilon$ yields

$$\int_0^\epsilon \int_{\partial B_R(\mathbf{x}_0)} \mathbf{v}_t^{\mathbf{x}_0, \mathbf{x}_T}(\mathbf{x}) \, p_t^{\mathbf{x}_0, \mathbf{x}_T}(\mathbf{x}) \cdot \mathbf{n} \, dS \, dt = \int_{B_R(\mathbf{x}_0)} p_{t=0}^{\mathbf{x}_0, \mathbf{x}_T}(\mathbf{x}) \, d\mathbf{x} - \int_{B_R(\mathbf{x}_0)} p_{t=\epsilon}^{\mathbf{x}_0, \mathbf{x}_T}(\mathbf{x}) \, d\mathbf{x}.$$

Since $p_{t=0}^{\mathbf{x}_0, \mathbf{x}_T} = \delta_{\mathbf{x}_0}$ and $\mathbf{x}_0 \in B_R(\mathbf{x}_0)$, the first integral equals 1. Therefore the lateral flux equals

$$\int_{\partial B_R \times [-\epsilon, \epsilon]} \mathbf{E}^{\mathbf{x}_0, \mathbf{x}_T} \cdot d\mathbf{S} = \int_0^\epsilon \int_{\partial B_R} \mathbf{v}_t^{\mathbf{x}_0, \mathbf{x}_T} \, p_t^{\mathbf{x}_0, \mathbf{x}_T} \cdot \mathbf{n} \, dS \, dt = 1 - \int_{B_R} p_{t=\epsilon}^{\mathbf{x}_0, \mathbf{x}_T}(\mathbf{x}) \, d\mathbf{x}.$$

*Top.* Finally, for the top part

$$\int_{B_R \times \{\epsilon\}} \mathbf{E}^{\mathbf{x}_0, \mathbf{x}_T} \cdot d\mathbf{S} = \int_{B_R} \mathbf{E}^{\mathbf{x}_0, \mathbf{x}_T}(\mathbf{x}, \epsilon)_t \, d\mathbf{x} = \int_{B_R} p_{t=\epsilon}^{\mathbf{x}_0, \mathbf{x}_T}(\mathbf{x}) \, d\mathbf{x}.$$

As a result, adding the fluxes through the bottom, lateral, and top part gives

$$\int_{\partial P} \mathbf{E}^{\mathbf{x}_0, \mathbf{x}_T} \cdot d\mathbf{S} = 0 + \left(1 - \int_{B_R} p_{t=\epsilon}^{\mathbf{x}_0, \mathbf{x}_T} \, d\mathbf{x}\right) + \int_{B_R} p_{t=\epsilon}^{\mathbf{x}_0, \mathbf{x}_T} \, d\mathbf{x} = 1.$$

Therefore $\int_{\partial M} \mathbf{E}^{\mathbf{x}_0, \mathbf{x}_T} \cdot d\mathbf{S} = 1$ for every closed surface $\partial M$ enclosing $\tilde{\mathbf{x}}_0$ but not $\tilde{\mathbf{x}}_T$. Hence the total flux is fixed and equals $\Phi_0 = 1$, independent of $(\mathbf{x}_0, \mathbf{x}_T)$.

This proves that $\mathbf{E}^{\mathbf{x}_0, \mathbf{x}_T}$ is a forward-only Interaction Field with unit total flux.

2. **Recovering the global field $\mathbf{E}(\mathbf{x}, t)$.**

By the generalized superposition principle,

$$\mathbf{E}(\mathbf{x}, t) = \int \mathbf{E}^{\mathbf{x}_0, \mathbf{x}_T}(\mathbf{x}, t) \, d\pi_{0, T}(\mathbf{x}_0, \mathbf{x}_T).$$

Taking components and using the definition (22) gives

$$\mathbf{E}(\mathbf{x}, t)_t = \int p_t^{\mathbf{x}_0, \mathbf{x}_T}(\mathbf{x}) \, d\pi_{0,T}(\mathbf{x}_0, \mathbf{x}_T) = p_t(\mathbf{x}),$$

where the last equality is exactly the marginalization formula for $p_t$. Similarly,

$$\mathbf{E}(\mathbf{x}, t)_{\mathbf{x}} = \int \mathbf{v}_t^{\mathbf{x}_0, \mathbf{x}_T}(\mathbf{x}) \, p_t^{\mathbf{x}_0, \mathbf{x}_T}(\mathbf{x}) \, d\pi_{0,T}(\mathbf{x}_0, \mathbf{x}_T).$$

By the definition of the CFM global velocity $\mathbf{v}_t(\mathbf{x})$ (conditional expectation given $x_t = \mathbf{x}$),

$$\mathbf{v}_t(\mathbf{x}) = \frac{\int \mathbf{v}_t^{\mathbf{x}_0, \mathbf{x}_T}(\mathbf{x}) \, p_t^{\mathbf{x}_0, \mathbf{x}_T}(\mathbf{x}) \, d\pi_{0,T}(\mathbf{x}_0, \mathbf{x}_T)}{\int p_t^{\mathbf{x}_0, \mathbf{x}_T}(\mathbf{x}) \, d\pi_{0,T}(\mathbf{x}_0, \mathbf{x}_T)} = \frac{\mathbf{E}(\mathbf{x}, t)_{\mathbf{x}}}{p_t(\mathbf{x})}.$$

Therefore $\mathbf{E}(\mathbf{x}, t)_{\mathbf{x}} = \mathbf{v}_t(\mathbf{x}) p_t(\mathbf{x})$, and hence

$$\mathbf{E}(\mathbf{x}, t) = \big(\mathbf{v}_t(\mathbf{x}) \, p_t(\mathbf{x}), \, p_t(\mathbf{x})\big).$$

3. **Field-line dynamics coincide with the CFM dynamics.**

For a forward-only global field, the data-space dynamics obtained by reparameterizing field lines by $t$ is

$$\frac{d\mathbf{x}}{dt} = \frac{\mathbf{E}(\mathbf{x}, t)_{\mathbf{x}}}{\mathbf{E}(\mathbf{x}, t)_t},$$

Using Item **2**,

$$\frac{\mathbf{E}(\mathbf{x}, t)_{\mathbf{x}}}{\mathbf{E}(\mathbf{x}, t)_t} = \frac{\mathbf{v}_t(\mathbf{x}) p_t(\mathbf{x})}{p_t(\mathbf{x})} = \mathbf{v}_t(\mathbf{x}),$$

so the induced dynamics coincide with the CFM generative ODE:

$$\frac{d\mathbf{x}}{dt} = \mathbf{v}_t(\mathbf{x}) = \frac{\mathbf{E}(\mathbf{x}, t)_{\mathbf{x}}}{\mathbf{E}(\mathbf{x}, t)_t}.$$

$\square$

*Remark* B.1. The locally Lipschitz condition on the conditional velocity field $\mathbf{v}_t^{\mathbf{x}_0, \mathbf{x}_T}(\mathbf{x})$ is essential for the proof of Theorem 3.1. It ensures the existence of a unique solution to the ODE $\frac{d\mathbf{x}}{dt} = \mathbf{v}_t^{\mathbf{x}_0, \mathbf{x}_T}(\mathbf{x})$ with prescribed endpoints, which in turn guarantees that the constructed field lines connect $\tilde{\mathbf{x}}_0$ to $\tilde{\mathbf{x}}_T$ without ambiguity. Without this regularity, the duality between conditional flows and interaction fields may break down due to non-unique or ill-defined trajectories.

### B.3. Assumptions for Theorem 3.3

Assume $\mathbf{E}^{\mathbf{x}_0, \mathbf{x}_T}(\mathbf{x}, t)$ is a forward-only Interaction Field with total flux $\Phi_0$, satisfying:

1. $\mathbf{E}^{\mathbf{x}_0, \mathbf{x}_T} \in C^1(\mathbb{R}^D \times (0, T))$, $\mathbf{E}_t^{\mathbf{x}_0, \mathbf{x}_T}(\mathbf{x}, t) > 0$ for $t \in (0, T)$ and $\mathbf{E}_t^{\mathbf{x}_0, \mathbf{x}_T}(\mathbf{x}, t) = 0$ for $t < 0$;

2. The measures converge $\lim_{t \to 0} \mathbf{E}_t^{\mathbf{x}_0, \mathbf{x}_T}(\mathbf{x}, t) \to \Phi_0 \delta_{\mathbf{x}_0}(\mathbf{x})$, $\lim_{t \to T} \mathbf{E}_t^{\mathbf{x}_0, \mathbf{x}_T}(\mathbf{x}, t) = \Phi_0 \delta_{\mathbf{x}_T}(\mathbf{x})$ in the weak sense;

3. $\mathbf{E}_{\mathbf{x}}^{\mathbf{x}_0, \mathbf{x}_T} / \mathbf{E}_t^{\mathbf{x}_0, \mathbf{x}_T}$ is locally Lipschitz in $\mathbf{x}$, uniformly in $t$ on compact subsets of $(0, T) \times \mathbb{R}^D$;

4. The Field decays sufficiently at infinity so that integrals converge ($\int_{\mathbb{R}^D} |\mathbf{E}_t^{\mathbf{x}_0, \mathbf{x}_T}(\mathbf{x}, t)| d\mathbf{x} < \infty$) and the flux through infinite lateral surfaces vanishes:

$$\lim_{R \to \infty} \int_{\partial B_R \times [t_1, t_2]} \mathbf{E}^{\mathbf{x}_0, \mathbf{x}_T}(\mathbf{x}, t) \cdot d\mathbf{S} = 0.$$

### B.4. Proof of Proposition 3.2.

Fix any $t^* \in (0, T)$. We show that the slice flux equals the total flux:

$$\int_{\mathbb{R}^D} \mathbf{E}^{\mathbf{x}_0, \mathbf{x}_T}(\mathbf{x}, t^*)_t \, d\mathbf{x} = \Phi_0. \tag{36}$$

First, for $R > 0$, define the cylinder

$$M_R := B_R(\mathbf{x}_0) \times [-t^*, t^*], \qquad B_R(\mathbf{x}_0) := \{\mathbf{x} \in \mathbb{R}^D : \|\mathbf{x} - \mathbf{x}_0\|_2 \leq R\}.$$

Since $t^* < T$, the volume $M_R$ encloses $\tilde{\mathbf{x}}_0 = (\mathbf{x}_0, 0)$ but not $\tilde{\mathbf{x}}_T = (\mathbf{x}_T, T)$. Hence, by the defining total-flux property of a forward-only Interaction Field,

$$\int_{\partial M_R} \mathbf{E}^{\mathbf{x}_0, \mathbf{x}_T} \cdot d\mathbf{S} = \Phi_0.$$

Next, let's decompose the boundary into bottom, lateral, and top parts:

$$\partial M_R = \underbrace{B_R \times \{-t^*\}}_{\text{bottom}} \cup \underbrace{\partial B_R \times [-t^*, t^*]}_{\text{lateral}} \cup \underbrace{B_R \times \{t^*\}}_{\text{top}}.$$

Using outward normals, we obtain

$$\int_{\partial M_R} \mathbf{E}^{\mathbf{x}_0, \mathbf{x}_T} \cdot d\mathbf{S} = -\int_{B_R} \mathbf{E}^{\mathbf{x}_0, \mathbf{x}_T}(\mathbf{x}, -t^*)_t \, d\mathbf{x} + \int_{\partial B_R \times [-t^*, t^*]} \mathbf{E}^{\mathbf{x}_0, \mathbf{x}_T} \cdot d\mathbf{S} + \int_{B_R} \mathbf{E}^{\mathbf{x}_0, \mathbf{x}_T}(\mathbf{x}, t^*)_t \, d\mathbf{x}. \quad (37)$$

The bottom term vanishes because (by the forward-only convention) $\mathbf{E}^{\mathbf{x}_0, \mathbf{x}_T}(\mathbf{x}, t) = \mathbf{0}$ for $t < 0$:

$$\int_{B_R} \mathbf{E}^{\mathbf{x}_0, \mathbf{x}_T}(\mathbf{x}, -t^*)_t \, d\mathbf{x} = 0.$$

Use the assumption that the field has vanishing lateral flux at infinity (Manukhov et al., 2026):

$$\lim_{R \to \infty} \int_{\partial B_R \times [-t^*, t^*]} \mathbf{E}^{\mathbf{x}_0, \mathbf{x}_T} \cdot d\mathbf{S} = 0.$$

Letting $R \to \infty$ in (37) therefore yields

$$\Phi_0 = \lim_{R \to \infty} \int_{\partial M_R} \mathbf{E}^{\mathbf{x}_0, \mathbf{x}_T} \cdot d\mathbf{S} = \lim_{R \to \infty} \int_{B_R} \mathbf{E}^{\mathbf{x}_0, \mathbf{x}_T}(\mathbf{x}, t^*)_t \, d\mathbf{x} = \int_{\mathbb{R}^D} \mathbf{E}^{\mathbf{x}_0, \mathbf{x}_T}(\mathbf{x}, t^*)_t \, d\mathbf{x},$$

which proves (36). $\qquad \square$

### B.5. Proof of Theorem 3.3

We prove the three items in order.

1. **The pair $(\mathbf{v}_t^{\mathbf{x}_0, \mathbf{x}_T}(\mathbf{x}), p_t^{\mathbf{x}_0, \mathbf{x}_T}(\mathbf{x}))$ defines a Conditional Flow.**

   *Well-definedness.* Since $\mathbf{E}^{\mathbf{x}_0, \mathbf{x}_T}$ is forward-only, $\mathbf{E}^{\mathbf{x}_0, \mathbf{x}_T}(\mathbf{x}, t)_t > 0$ wherever the field is defined. Hence $\mathbf{v}_t^{\mathbf{x}_0, \mathbf{x}_T}(\mathbf{x})$ is well-defined and $p_t^{\mathbf{x}_0, \mathbf{x}_T}(\mathbf{x}) \geq 0$.

   *Normalization.* By Proposition 3.2, for every $t \in (0, T)$,

   $$\int_{\mathbb{R}^D} \mathbf{E}^{\mathbf{x}_0, \mathbf{x}_T}(\mathbf{x}, t)_t \, d\mathbf{x} = \Phi_0.$$

   Therefore,

   $$\int_{\mathbb{R}^D} p_t^{\mathbf{x}_0, \mathbf{x}_T}(\mathbf{x}) \, d\mathbf{x} = \frac{1}{\Phi_0} \int_{\mathbb{R}^D} \mathbf{E}^{\mathbf{x}_0, \mathbf{x}_T}(\mathbf{x}, t)_t \, d\mathbf{x} = 1.$$

   *Continuity equation.* Using $p_t^{\mathbf{x}_0, \mathbf{x}_T}(\mathbf{x}) = \mathbf{E}^{\mathbf{x}_0, \mathbf{x}_T}(\mathbf{x}, t)_t / \Phi_0$ and $\mathbf{v}_t^{\mathbf{x}_0, \mathbf{x}_T}(\mathbf{x}) \, p_t^{\mathbf{x}_0, \mathbf{x}_T}(\mathbf{x}) = \mathbf{E}^{\mathbf{x}_0, \mathbf{x}_T}(\mathbf{x}, t)_{\mathbf{x}} / \Phi_0$, we obtain

   $$\partial_t p_t^{\mathbf{x}_0, \mathbf{x}_T}(\mathbf{x}) + \nabla_{\mathbf{x}} \cdot \left( \mathbf{v}_t^{\mathbf{x}_0, \mathbf{x}_T}(\mathbf{x}) \, p_t^{\mathbf{x}_0, \mathbf{x}_T}(\mathbf{x}) \right) = \frac{1}{\Phi_0} \left( \partial_t \mathbf{E}^{\mathbf{x}_0, \mathbf{x}_T}(\mathbf{x}, t)_t + \nabla_{\mathbf{x}} \cdot \mathbf{E}^{\mathbf{x}_0, \mathbf{x}_T}(\mathbf{x}, t)_{\mathbf{x}} \right)$$
   $$= \frac{1}{\Phi_0} \nabla_{(\mathbf{x}, t)} \cdot \mathbf{E}^{\mathbf{x}_0, \mathbf{x}_T}(\mathbf{x}, t) = 0,$$

   where the last equality is the divergence-free property of Interaction Fields.

   *Boundary conditions.* In the IFM capacitor setup, the conditional field has only the endpoint source/sink on the plates. Equivalently, the plate flux is concentrated at the endpoints (suggested to be stated as part of the forward-only IFM definition / standing assumptions): in the sense of measures (distributions) on $\mathbb{R}^D$,

   $$\mathbf{E}^{\mathbf{x}_0, \mathbf{x}_T}(\mathbf{x}, 0)_t = \Phi_0 \, \delta_{\mathbf{x}_0}(\mathbf{x}), \qquad \mathbf{E}^{\mathbf{x}_0, \mathbf{x}_T}(\mathbf{x}, T)_t = \Phi_0 \, \delta_{\mathbf{x}_T}(\mathbf{x}).$$

   Dividing by $\Phi_0$ gives

   $$p_{t=0}^{\mathbf{x}_0, \mathbf{x}_T}(\mathbf{x}) = \delta_{\mathbf{x}_0}(\mathbf{x}), \qquad p_{t=T}^{\mathbf{x}_0, \mathbf{x}_T}(\mathbf{x}) = \delta_{\mathbf{x}_T}(\mathbf{x}),$$

which is exactly (2). Thus $(\mathbf{v}_t^{\mathbf{x}_0,\mathbf{x}_T}(\mathbf{x}), p_t^{\mathbf{x}_0,\mathbf{x}_T}(\mathbf{x}))$ is a Conditional Flow.

2. **Recovering $(\mathbf{v}_t(\mathbf{x}), p_t(\mathbf{x}))$ from the global field $\mathbf{E}(\mathbf{x}, t)$.**

By generalized superposition (9),

$$\mathbf{E}(\mathbf{x}, t) \;=\; \int \mathbf{E}^{\mathbf{x}_0,\mathbf{x}_T}(\mathbf{x}, t)\, \pi_{0,T}(\mathbf{x}_0, \mathbf{x}_T)\, d\mathbf{x}_0\, d\mathbf{x}_T.$$

Taking the $t$-component and using $p_t^{\mathbf{x}_0,\mathbf{x}_T} = \mathbf{E}^{\mathbf{x}_0,\mathbf{x}_T}{}_t/\Phi_0$ gives

$$\mathbf{E}(\mathbf{x}, t)_t = \int \mathbf{E}^{\mathbf{x}_0,\mathbf{x}_T}(\mathbf{x}, t)_t\, \pi_{0,T}\, d\mathbf{x}_0\, d\mathbf{x}_T = \Phi_0 \int p_t^{\mathbf{x}_0,\mathbf{x}_T}(\mathbf{x})\, \pi_{0,T}\, d\mathbf{x}_0\, d\mathbf{x}_T.$$

By the CFM marginalization formula (3), $p_t(\mathbf{x}) = \int p_t^{\mathbf{x}_0,\mathbf{x}_T}(\mathbf{x})\, \pi_{0,T}\, d\mathbf{x}_0\, d\mathbf{x}_T$, hence

$$p_t(\mathbf{x}) = \frac{\mathbf{E}(\mathbf{x}, t)_t}{\Phi_0}.$$

Similarly, using $\mathbf{v}_t^{\mathbf{x}_0,\mathbf{x}_T}(\mathbf{x})\, p_t^{\mathbf{x}_0,\mathbf{x}_T}(\mathbf{x}) = \mathbf{E}^{\mathbf{x}_0,\mathbf{x}_T}(\mathbf{x}, t)_{\mathbf{x}}/\Phi_0$,

$$\mathbf{E}(\mathbf{x}, t)_{\mathbf{x}} = \int \mathbf{E}^{\mathbf{x}_0,\mathbf{x}_T}(\mathbf{x}, t)_{\mathbf{x}}\, \pi_{0,T}\, d\mathbf{x}_0\, d\mathbf{x}_T = \Phi_0 \int \mathbf{v}_t^{\mathbf{x}_0,\mathbf{x}_T}(\mathbf{x})\, p_t^{\mathbf{x}_0,\mathbf{x}_T}(\mathbf{x})\, \pi_{0,T}\, d\mathbf{x}_0\, d\mathbf{x}_T.$$

On the other hand, by the definition of the global drift (4),

$$\mathbf{v}_t(\mathbf{x})\, p_t(\mathbf{x}) = \mathbf{E}_{\mathbf{x}_0,\mathbf{x}_T}\left[\mathbf{v}_t^{\mathbf{x}_0,\mathbf{x}_T}(\mathbf{x}_t) \mid \mathbf{x}_t = \mathbf{x}\right] p_t(\mathbf{x})$$

$$= p_t(\mathbf{x}) \int \mathbf{v}_t^{\mathbf{x}_0,\mathbf{x}_T}(\mathbf{x})\, p_t(\mathbf{x}_0, \mathbf{x}_T \mid \mathbf{x}_t = \mathbf{x})\, d\mathbf{x}_0\, d\mathbf{x}_T$$

$$= p_t(\mathbf{x}) \int \mathbf{v}_t^{\mathbf{x}_0,\mathbf{x}_T}(\mathbf{x})\, \frac{p_t^{\mathbf{x}_0,\mathbf{x}_T}(\mathbf{x})\, \pi_{0,T}(\mathbf{x}_0, \mathbf{x}_T)}{p_t(\mathbf{x})}\, d\mathbf{x}_0\, d\mathbf{x}_T$$

$$= \int \mathbf{v}_t^{\mathbf{x}_0,\mathbf{x}_T}(\mathbf{x})\, p_t^{\mathbf{x}_0,\mathbf{x}_T}(\mathbf{x})\, \pi_{0,T}(\mathbf{x}_0, \mathbf{x}_T)\, d\mathbf{x}_0\, d\mathbf{x}_T.$$

Combining the last three displays yields

$$\mathbf{v}_t(\mathbf{x})\, p_t(\mathbf{x}) = \frac{\mathbf{E}(\mathbf{x}, t)_{\mathbf{x}}}{\Phi_0}.$$

Substituting $p_t(\mathbf{x}) = \mathbf{E}(\mathbf{x}, t)_t/\Phi_0$ gives

$$\mathbf{v}_t(\mathbf{x}) = \frac{\mathbf{E}(\mathbf{x}, t)_{\mathbf{x}}}{\mathbf{E}(\mathbf{x}, t)_t},$$

proving (28).

3. **Flow-based dynamics coincide with global field lines.**

Let $\tilde{\mathbf{x}}(\tau) = (\mathbf{x}(\tau), t(\tau))$ be a field line of the global field:

$$\frac{d\tilde{\mathbf{x}}(\tau)}{d\tau} = \mathbf{E}(\tilde{\mathbf{x}}(\tau)).$$

Since $\mathbf{E}$ is a nonnegative superposition of forward-only fields, $\mathbf{E}_t(\mathbf{x}, t) \geq 0$ everywhere (and $\mathbf{E}_t > 0$ along any nontrivial field line). Hence $\frac{dt}{d\tau} = \mathbf{E}_t(\tilde{\mathbf{x}}(\tau)) > 0$ and we may reparameterize the curve by $t$:

$$\frac{d\mathbf{x}}{dt} = \frac{d\mathbf{x}/d\tau}{dt/d\tau} = \frac{\mathbf{E}(\mathbf{x}, t)_{\mathbf{x}}}{\mathbf{E}(\mathbf{x}, t)_t} = \mathbf{v}_t(\mathbf{x}),$$

where the last equality follows from item **2**. This is exactly the CFM generative dynamics (4).

$\square$

## C. Additional Experiments Details

We built our implementation on top of the official repository for PFGM++ (Xu et al., 2023)[1]. We used the same hyperparameters and model architecture, except for the training batch size, which in our case was $B = 256$. We trained our final models

---

[1] https://github.com/Newbeeer/pfgmpp

for 100k kimg; here, kimg denotes the number of images (in thousands) that were passed through the model during training. For multi-sample target computation, we constructed a batch to compute a multi-sample target of size $N \geq B$ (for $N > 1$), which is a concatenation of $B$ samples that we compute the loss on and additional random $N - B$ samples drawn from the dataset. For every FID (Heusel et al., 2018) evaluation, we performed sampling with 50 steps of the Euler sampler for the two-sided interpolant and the Heun sampler for EDM and PFGM++. We measure FID between 50k generated samples and samples from the dataset that the model was trained on.

**Two-sided Interpolant.** For the two-sided interpolant, we used the following definition of $\mathbf{x}_t$ from (Albergo & Vanden-Eijnden, 2022):

$$\mathbf{x}_t = (1 - t) \cdot \mathbf{x}_0 + t \cdot \mathbf{x}_T + C \cdot \sqrt{t(t-1)} \cdot \varepsilon, \ \varepsilon \sim \mathcal{N}(0, I), \tag{38}$$

where the prefactor $C$ is a hyperparameter that scales the variance of the random noise in the stochastic interpolant. The original paper includes an additional $\sqrt{2}$ prefactor for $\varepsilon$; to account for this, we define $s := C/\sqrt{2}$. We conducted experiments that reveal the dependency of generation quality on the parameter $s$. In Figure 3, it can be seen that larger values of $s$ lead to higher FID scores, and the best value of $s$ in terms of quality was determined to be $s = 0.1$.

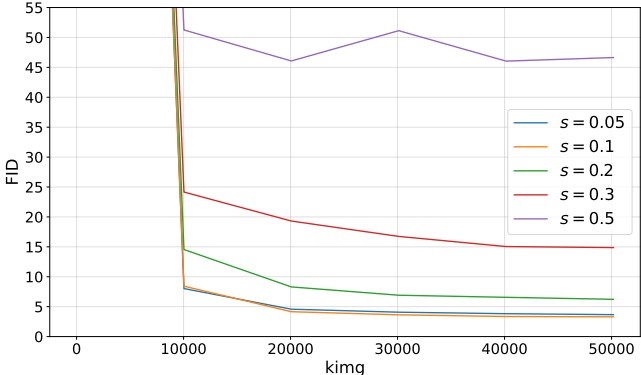

*Figure 3.* FID for the two-sided interpolant with different values of the hyperparameter $s$ during training. Even a small increase of $s$ to 0.2 leads to FID values that are too high to consider using this model.

The model for the two-sided interpolant is trained to approximate the conditional velocity $\mathbf{v}_t^{\mathbf{x}_0, \mathbf{x}_T}$, as stated in Table 3. For the one-sided interpolant, we followed the EDM and PFGM++ implementations and trained the model to approximate the original data sample $\mathbf{x}_0$.

**Multi-Sample Target.** For two-sided IFM, we used the velocity formulation and trained our model to approximate the multi-sample target $\mathbf{v}_t^N$ from the corresponding formula. For EDM and PFGM++, the formula for the conditional velocity field is shown in Table 3. It is possible to construct a multi-sample velocity estimate $\mathbf{v}_\sigma^N(\mathbf{x}_\sigma)$ using formula (31) for both EDM and PFGM++. To do so, we use $p_\sigma(\mathbf{x}|\mathbf{x}_0^i)$ to compute $w_i$ for each $\mathbf{x}_0^i \sim \pi_0, i \in \{1, \ldots, N\}$.

Using the multi-sample velocity target, we can derive a multi-sample $\mathbf{x}_0$ target, which we denote as $\mathbf{x}_0^N(\mathbf{x}_\sigma)$. This allows us to keep the same hyperparameters and sampling algorithm as in EDM and PFGM++. We can write the multi-sample estimate of the $\mathbf{x}_0$ target as:

$$\mathbf{x}_0^N(\mathbf{x}_\sigma) = \mathbf{x}_\sigma - \sigma \cdot \mathbf{v}_\sigma^N(\mathbf{x}_\sigma) = \mathbf{x}_\sigma - \sigma \cdot \frac{\sum_{i=1}^N \mathbf{v}_\sigma^{\mathbf{x}_0}(\mathbf{x}_\sigma) p_\sigma(\mathbf{x}_\sigma|\mathbf{x}_0^i)}{\sum_{i=1}^N p_\sigma(\mathbf{x}_\sigma|\mathbf{x}_0^i)} = \mathbf{x}_\sigma - \frac{\sum_{i=1}^N (\mathbf{x}_\sigma - \mathbf{x}_0^i) p_\sigma(\mathbf{x}_\sigma|\mathbf{x}_0^i)}{\sum_{i=1}^N p_\sigma(\mathbf{x}_\sigma|\mathbf{x}_0^i)}, \tag{39}$$

where $\mathbf{x}_0^i \sim \pi_0, i \in \{1, \ldots, N\}$.

**Training Plots.** In our work, we investigated the dependency of generation quality (in terms of FID) on the number of samples used to estimate the multi-sample target (see formula (31)). We denote the number of samples used for target estimation as $N$ in this section. Plots of training convergence with different values of $N$ in terms of FID (Heusel et al., 2018) for the two-sided interpolant from equation (38), as well as for EDM and PFGM++ with $d \in \{128, 2048\}$, are shown in Figure 4.

**Our Code.** We provide the code in archive as supplementary material. All results from our paper can be reproduced with it.

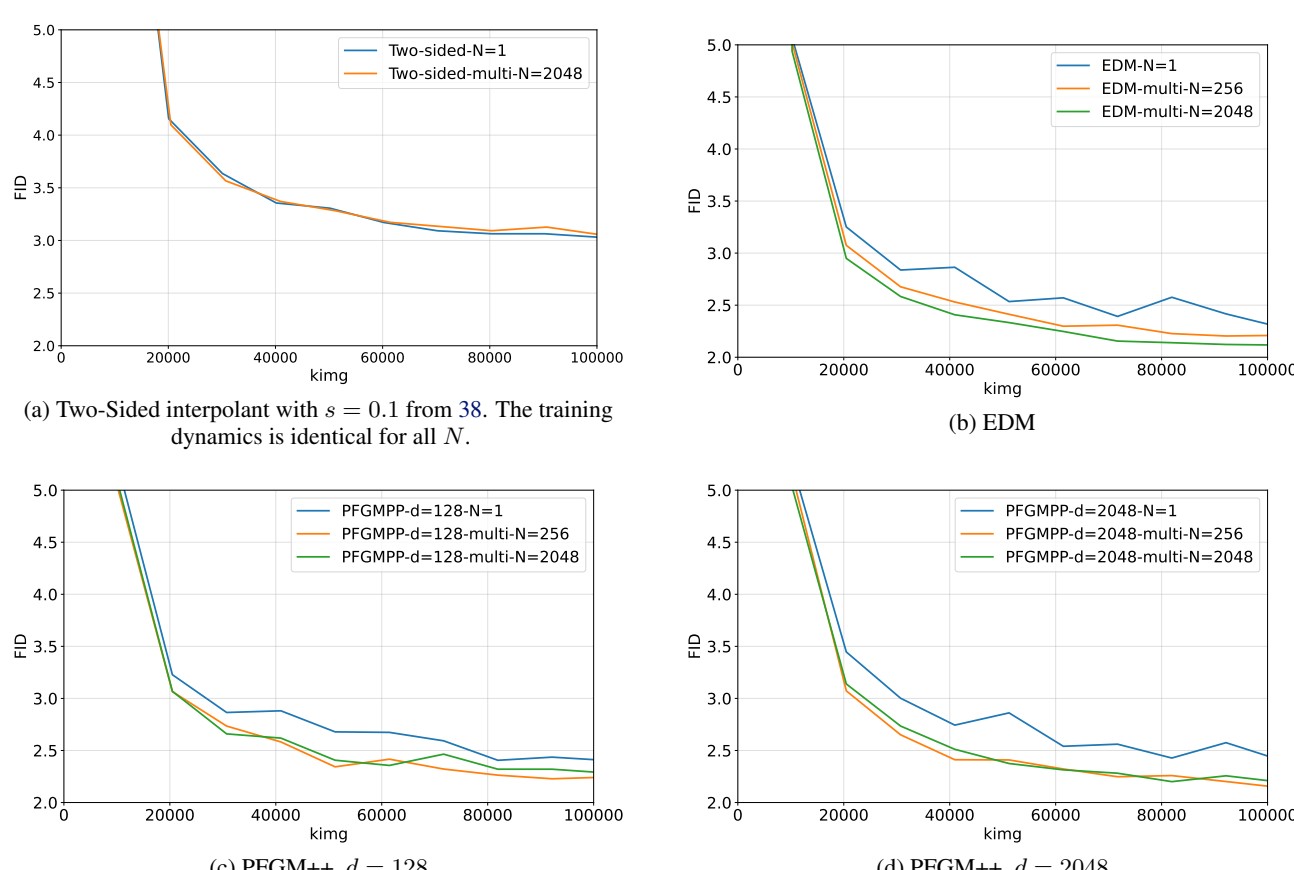

(a) Two-Sided interpolant with $s = 0.1$ from [38]. The training dynamics is identical for all $N$.

(b) EDM

(c) PFGM++, $d = 128$

(d) PFGM++, $d = 2048$

*Figure 4.* Dependence of FID from kimg of training for different interpolants.

