# OpenReview forum: "Unlocking the Duality between Flow and Field Matching"
_ICML.cc/2026/Conference — Submitted to ICML 2026_

### Official Review · Reviewer_yqaY · 2026-02-25

**Soundness:** 2
**Presentation:** 3
**Significance:** 2
**Originality:** 3
**Overall Recommendation:** 4
**Confidence:** 3

**Summary:**

This paper shows that CFM induces the same generative dynamics as a natural subclass of IFM by constructing a bijection between the two formulations: it maps conditional flows to forward-only interaction fields, and conversely maps forward-only interaction fields back to conditional flows. Building on this result, the paper argues that the duality provides a probabilistic interpretation of forward-only IFM and motivates new CFM techniques inspired by the IFM perspective.

**Compliance With Llm Reviewing Policy:**

Affirmed.

**Final Justification:**

The rebuttal addressed your main concerns. I recommend the acceptance of this paper.

**Key Questions For Authors:**

- Is this assumption mainly a technical requirement for the proof, or is it fundamentally necessary for the equivalence between CFM and IFM?
- In practice, do neural parameterizations used in modern generative models satisfy these assumptions approximately?

**Limitations:**

- The empirical evaluation is limited to CIFAR-10, which restricts the strength of the evidence for the practical implications of the theoretical results.

- The paper does not sufficiently clarify the assumptions underlying the main theoretical results, which makes it harder to assess their scope and applicability.

**Strengths And Weaknesses:**

**Strengths:**

- The paper provides a unified perspective on CFM and IFM, which is conceptually elegant and of clear theoretical interest.

- The theoretical results offer useful insights for practical training of generative models (e.g., by motivating IFM-inspired techniques for CFM).

- The paper includes a helpful overview of related generative modeling frameworks, which improves readability and makes the presentation easier to follow.

**Weaknesses:**

- The empirical evaluation is somewhat limited and does not yet provide strong evidence for practical impact. In particular, the experiments are conducted on a limited set of datasets.

- The main theoretical results rely on several “mild” assumptions, but their scope and limitations are not sufficiently discussed in the main text. A clearer discussion would improve transparency and help readers better assess the applicability of the theory.

---

> ### Author Rebuttal · Authors · 2026-03-31
>
> **Q1 + W2: Is this assumption mainly a technical requirement for the proof, or is it fundamentally necessary for the equivalence between CFM and IFM?**
>
> Regarding the assumptions, we agree that the current draft does not separate clearly enough what is *structural* versus what is mainly *technical*. We will revise the paper to make this distinction explicit. More specifically:
>
> - The **structural assumptions** are the ones that define the exact regime of equivalence. In particular, the key structural condition is the forward-only requirement $E_t^{x_0, x_T}(\mathbf x,t)>0$ on $(0,T)$: this is precisely what allows IFM field lines to be identified with forward time-parameterized trajectories, and hence with CFM. Likewise, endpoint matching is structural at the level of the conditional construction.
> - By contrast, $C^1$ regularity, convergence of the relevant integrals, decay at infinity, and the precise local-Lipschitz assumptions are mainly **technical conditions** used to make the theorem statement and proof clean. The local-Lipschitz assumptions do play a structural role in ensuring a well-defined ODE/trajectory interpretation, but their exact form is a convenient sufficient condition rather than a fundamental necessity.
>
> **Q2: In practice, do neural parameterizations used in modern generative models satisfy these assumptions approximately?**
>
> In generative models, these assumptions are typically satisfied only approximately rather than exactly. Our view is that the theory characterizes the *idealized continuous objects* that neural networks are trained to approximate.
>
> Commonly used neural vector fields are continuous / piecewise smooth and are intended to approximate the regular forward field regime covered by the theory, even though finite-capacity networks and finite training error do not enforce the assumptions pointwise. Standard neural vector fields with common activations are also typically locally Lipschitz in $x$, although they may fail to be globally $C^1$ unless smooth activations are used. Strict positivity of $p_t^{x_0, x_T}$ or $E_t^{x_0, x_T}$ is not automatic for arbitrary architectures, but it can be approximately enforced by construction, e.g., by parameterizing the time component through a positive output. Exact weak convergence to endpoint Dirac masses and exact vanishing flux at infinity should be viewed as idealized assumptions on the target continuous objects rather than literal properties of finite neural networks.
>
> We will add a discussion clarifying this “theory for the target object, approximation in practice” interpretation.
>
> **W1: The empirical evaluation is somewhat limited and does not yet provide strong evidence for practical impact. In particular, the experiments are conducted on a limited set of datasets.**
>
> The goal of section 4 is to show that, although IFM-derived techniques are promising, they remain difficult to use in high dimensions because constructing a low-variance multi-sample target remains challenging, as does the corresponding Gini-based analysis.
>
> To better highlight these challenges, we provide additional evaluation on the higher-dimensional FFHQ64 dataset and a Gini analysis across datasets of different dimensionalities (see [*, $\S$1 and $\S$3] for details). The results suggest that the difficulty grows with dimension: FFHQ 64×64 shows less benefit than CIFAR 32×32, and the weights become more degenerate as dimensionality increases.
>
> That said, we agree that we should also demonstrate the potential practical benefits of these techniques. To this end, we add two 2D toy experiments. First, we show the importance of theory-induced volume coverage for IFM in [\*, $\S$5]. Second, we show that the multi-sample target has lower variance and enables faster convergence in [\*, $\S$6].
>
> We will add these experiments in final version.
>
> **Conclusion remarks:** We hope these clarifications have fully addressed your concerns. Should any points require further elaboration, we are happy to provide it. Based on these comprehensive explanations, we are hopeful that you will consider raising your rating.
>
> [*] https://anonym.link/?https://zenodo.org/records/19347539

---

> > ### Author Rebuttal · Reviewer_yqaY · 2026-04-02
> >
> > My concerns have been fully resolved.

---

> > > ### Author Response · Authors · 2026-04-02
> > >
> > > Thank you very much for your careful reading and thoughtful feedback. We really appreciate the time and attention you gave to our paper and to the rebuttal.
> > >
> > > We are glad that our response clarified the main points and helped resolve your concerns. In the final version, we will make these points more explicit, in particular the distinction between structural and technical assumptions, how the theory should be interpreted in the context of practical neural parameterizations, and the additional experimental discussion.

---

### Official Review · Reviewer_7CSX · 2026-03-12

**Soundness:** 3
**Presentation:** 3
**Significance:** 2
**Originality:** 2
**Overall Recommendation:** 3
**Confidence:** 3

**Summary:**

The paper connects two frameworks for training generative models. Conditional flow matching (CFM) learns a velocity field that transports a noise distribution to a data distribution along continuous paths. Interaction field matching (IFM) instead learns a divergence-free vector field in an extended space. The paper shows that when IFM field lines only point forward in "time", the two frameworks are equivalent and gives explicit construction mapping one to the other. It also shows that general IFM is strictly more expressive than CFM, since it allows backward oriented field lines. As practical application the paper derives a multi sample velocity estimator and evaluates it on CIFAR-10.

**Compliance With Llm Reviewing Policy:**

Affirmed.

**Final Justification:**

The authors have been transparent and responsive throughout the rebuttal. The theoretical contribution (the duality itself) is clean and clearly stated. However, the practical implications remain limited. The multi-sample estimator degrades with dimensionality and the most promising applications (drifting models, volume coverage) are left to future work. The paper is a solid expository/unifying contribution but the significance does not clear the acceptance threshold in my view.

**Key Questions For Authors:**

1. The FID improvements from the multi-sample estimator in Table 2 are modest. Were these results averaged over multiple random seeds? It is difficult to assess whether these differences are meaningful.
2.  The Gini coefficient analysis is only conducted on CIFAR10. How does the weight concentration behave as data dimension increases?
3. Beyond the equivalence itself what practical benefit does the field viewpoint provide? The multi sample estimator shows limited gains and the volume coverage idea is left to future work.

**Limitations:**

yes

**Strengths And Weaknesses:**

**Strengths**


- The equivalence between forward only IFM and CFM is clearly stated and the explicit constructions in both directions are useful.
- The paper precisely characterizes where CFM and IFM diverge.
- Table 3 provides a handy reference of dual constructions for many known models.
- The one sided IFM formulation is a natural extension that recovers PFGM as a special case.


**Weaknesses**

- The main duality result is mathematically interesting, but its actual usefulness remains unclear. The paper shows that the two formulations are equivalent, but it does not convincingly demonstrate what this equivalence enables beyond a change of perspective.
- The expressiveness gap between IFM and CFM is noted but unexploited. The paper acknowledges that backward oriented field lines are "not straightforward to use in practice" but does not investigate further.
- The multi sample estimator seems orthogonal to the main contribution and its effectiveness appears limited. The FID improvements are modest at best and the Gini coefficient analysis shows the weights are highly concentrated on a single sample.
- The volume coverage distribution idea is left entirely to future work.

---

> ### Author Rebuttal · Authors · 2026-03-31
>
> ## Reviewer 7CSX
>
> **Q2+Q3+W1+W3: Practical benefits. Gini analysis for higher dimensions.**
>
> The goal of section 4 is to show that, although IFM-derived techniques are promising, they remain difficult to use in high dimensions because constructing a low-variance multi-sample target remains challenging, as does the corresponding Gini-based analysis.
>
> To better highlight these challenges, we provide additional evaluation on the higher-dimensional FFHQ64 dataset and a Gini analysis across datasets of different dimensionalities (see [*, $\S$1 and $\S$3] for details). The results suggest that the difficulty grows with dimension: FFHQ 64×64 shows less benefit than CIFAR 32×32, and the weights become more degenerate as dimensionality increases.
>
> That said, we agree that we should also demonstrate the potential practical benefits of these techniques. To this end, we add two 2D toy experiments. First, we show the importance of theory-induced volume coverage for IFM in [\*, $\S$5]. Second, we show that the multi-sample target has lower variance and enables faster convergence in [\*, $\S$6].
>
> We will add these experiments in the final version.
>
> **Q1: Means and stds. Improvement significance.**
>
> We provide the mean and standard deviation over different seeds for both CIFAR and the newly added FFHQ dataset at 64×64 resolution in [*, $\S$1 and $\S$2].
>
> While CIFAR still shows a significant boost from the multi-sample target, the improvement for FFHQ is less pronounced. These results are consistent with our Gini analysis and support the proposition that a naive multi-sample target becomes increasingly difficult to construct as dimensionality grows. We consider addressing this challenge a promising direction for future work.
>
> **W2: Potential benefits of IFM expressivity.**
>
> EFM has an expressive advantage that cannot be captured by CFM. As shown in Table 2 and Figure 2, the two-sided linear interpolant does not benefit from the multi-sample target. By contrast, the electrostatic field satisfies a superposition property, so the objective can be decomposed into one-sided fields induced by the source and target datasets:
>
> $$E(x) = arg\min_{f(x)} | f(x) - \mathbb{E}_{x_0, x_T} E^{x_0, x_T}(x) |_2^2$$
>
> $$=arg\min_{f(x)} | f(x) - \mathbb{E}_{x_0, x_T} ( E^{x_0}(x) - E^{x_T}(x) ) |_2^2$$
>
> $$=arg\min_{f(x)} \mathbb{E}_{x_0} | f(x) - E^{x_0}(x) |_2^2$$
>
> $$+ \mathbb{E}_{x\_T} | f(x) + E^{x\_T}(x) |_2^2$$
>
> Thus, the two-sided multi-sample target reduces to two one-sided targets, which are shown to work better.
>
> **W4: The volume coverage distribution idea is left entirely to future work.**
>
> As we state in the section, exploring volume coverage distributions is only feasible when the multi-sample target can be constructed reliably. At present, there are only two examples of heuristic volume coverage distributions, namely EFM and PFGM, both of which show reasonably good results.
>
> We will add add this examples in the final version.
>
> **Conclusion remarks:** We hope these clarifications have fully addressed your concerns. Should any points require further elaboration, we are happy to provide it. Based on these comprehensive explanations, we are hopeful that you will consider raising your rating.
>
> [*] https://anonym.link/?https://zenodo.org/records/19347539

---

> > ### Author Rebuttal · Reviewer_7CSX · 2026-04-04
> >
> > Thank you for the additional experiments and clarifications.
> >
> > The seed-averaged FID results (Tables 1-2) address my concern about statistical significance.
> >
> > However, the new Gini analysis across resolutions (Table 3) reinforces my concern. If I am not mistaken, the Gini drops from ~0.21 at 8x8 to ~0.001 at 64x64, meaning the estimator becomes effectively single-sample at higher resolutions. The variance reduction and convergence experiments (Tables 6-7) are on a 2D toy target, which is exactly the regime where the weights are spread out. The Gini analysis seems to provide direct evidence that the estimator degrades with dimensionality. I appreciate the authors transparency on this point but it leaves the main practical contribution without a clear path to higher dimensional settings. I will maintain my score.

---

> > > ### Author Response · Authors · 2026-04-05
> > >
> > > We agree that the added Gini analysis points to an important limitation of the naive multi-sample estimator: as dimensionality increases, the weights become highly concentrated, so the estimator becomes effectively dominated by very few samples. We will make this limitation more explicit in the final version.
> > >
> > > That said, we would like to clarify the intended scope of the paper. Our main contribution is theoretical: we establish a constructive duality between CFM and forward-only IFM, show that general IFM is strictly more expressive than CFM, and use this duality to give forward-only IFM a probabilistic interpretation. The multi-sample estimator is not the core theorem of the paper; rather, it is a **first practical consequence suggested by the duality**.
> > >
> > > We therefore agree with your conclusion about the current naive estimator: in higher-dimensional settings it does not yet provide a clear scalable solution. In our view, this is still a useful outcome of the paper, because it identifies the specific failure mode: dominant-sample behavior, and thereby clarifies what needs to be improved next. In particular, the paper already points to more informed pair-proposal strategies as a natural direction for reducing this degeneracy.
> > >
> > > So, to be precise, our claim is not that the current naive multi-sample target already solves high-dimensional training. Our claim is that the duality reveals this estimator naturally, gives a principled interpretation of it, and helps diagnose why the naive version breaks down. We will revise the wording to make this scope clearer.

---

### Official Review · Reviewer_NiHa · 2026-03-13

**Soundness:** 3
**Presentation:** 3
**Significance:** 3
**Originality:** 3
**Overall Recommendation:** 4
**Confidence:** 3

**Summary:**

The scope of this paper is a theoretical contribution, studying the relationship between Conditional Flow Matching (CFM) and the newer Interaction Field Matching (IFM) framework for generative modeling. It is established that Conditional Flow Matching (CFM) is equivalent to
a subclass of IMFs, called forward-only / one-sided view of IFM, and proves a constructive two-way correspondence between the two matching paradigms. The paper also argues that general IFM is strictly more expressive than standard CFM because it allows backward-oriented field lines (e.g., EFM), which lie outside the forward-only setting.

**Compliance With Llm Reviewing Policy:**

Affirmed.

**Final Justification:**

The rebuttal answered adequately most of my questions thus I am increasing my score to a 4. I think this paper is above the acceptance threshold.

**Key Questions For Authors:**

* Can the authors provide direct empirical verification of the duality itself on a toy problem by constructing both the CFM and forward-only IFM representations and showing that they induce the same trajectories/marginals numerically?

* How does the proposed multi-sample target compare empirically and conceptually to recent multi-sample target ideas for Flow Matching, beyond the brief discussion in the text? Can the authors provide a direct comparison, which would be important for isolating what is genuinely new here?

* It is claimed that the **global field is directly estimated by superposition over samples**, whereas in CFM the global drift is written as a conditional expectation that is “generally intractable.” Can the authors provide examples where this can be used to treat cases where the intractability of CFMs can not yield results?

**Limitations:**

No. The paper does acknowledge some limitations implicitly—for example, the multi-sample target gives only slight gains for EDM/PFGM++ and no gain for the two-sided interpolant.
A stronger discussion would explicitly state the scope of the theory (it applies to forward-only IFM, not full IFM), the limited practical validation, the computational overhead/benefit tradeoff of the multi-sample estimator, and plausible downstream risks of improved generative modeling, even if those risks are indirect or low-stakes.

**Strengths And Weaknesses:**

## Strengths
* Clear conceptual contribution. The paper gives a genuinely useful unification of two frameworks that are usually presented very differently.
* Theory is rigorous and main the main contribution is largely conceptual/unifying, and the underlying mapping is elegant.
* The paper is well written and does a good job disentangling three regimes: standard CFM, forward-only IFM, and general IFM; Table 1 and the boxed takeaways make this distinction much easier to follow.
---
## Weaknesses
* The empirical validation is narrow. The experiments evaluate only one practical implication of the theory, on CIFAR-10, and do not directly verify the claimed duality by constructing matched CFM and forward-only IFM dynamics and showing numerical agreement.

* The paper’s novelty is strongest at the theorem/unification level rather than at the algorithmic level. While the constructive equivalence appears nontrivial, the practical method proposed in Section 4 is modest and its gains are limited.

* The claim that general IFM is strictly more expressive than CFM is theoretically well motivated, but the paper does not establish when this additional expressiveness matters in practice.

---

> ### Author Rebuttal · Authors · 2026-03-31
>
> **Q1 + W1: Empirical Validation of the revealed duality.**
>
> We verify the duality on a 2D toy transport problem (see [*, $\S$4] for details). We define two conditional flows via stochastic interpolants, map them to forward-only interaction fields, and compare the induced drifts (4) and (12) on common space-time points from the transport region. This directly tests identity (24).
>
> Matched pairs have errors about an order of magnitude smaller than mismatched pairs. This is exactly what the duality predicts: for the same underlying construction, the CFM drift and the forward-only IFM-induced drift coincide, whereas different interpolants produce genuinely different dynamics. The remaining nonzero MSE in the matched pairs is attributable to numerical approximation and imperfect training.
>
> **Q2: Multi-sample target difference to recent Flow Matching ideas?**
>
> Given the global field estimator $E$, there are two types of multi-sample targets.
>
> - The first is $E_x/E_z\in\mathbb{R}^D$. This is the same target used in the cited Flow Matching work, provided that one uses the underlying conditional flow corresponding to Flow Matching.
>
> - The second is $E/\|E\|_2\in\mathbb{R}^{D+1}$. This is another multi-sample target used in PFGM and IFM,and it is novel relative to previous CFM works. Its main differences are (1) that it has unit norm, which may help training, and (2) that it requires the network output dimension to be one larger than the data dimension, requiring adaptation of standard U-Nets.
>
> **Q3: CFM drift intractability.**
>
> The CFM drift is generally intractable because it implicitly requires the marginals $p_t(x)$ for normalization. In standard CFM for training transport this is less problematic because the conditional expectation can be learned by regression. In drifting models [1], however, normalization remains difficult.
>
> More specifically, drifting models require a normalization constant estimated nonparametrically, which demands many samples per class. This is unrealistic for text-to-image tasks, where there is often only one image per class, i.e., per text prompt.
>
> One might instead try to train the drift in drifting models directly. However, the induced regression does not support training at arbitrary points (only at points induces by conditional distribution), which is important for drifting models. By contrast, in the electrostatic field formulation, the global field can be trained at any point $x$ using only a single target sample:
> $$E(x)=\arg\min_{f(x)}|f(x)-\mathbb{E}_{x_0,x_T}E^{x_0,x_T}(x)|_2^2$$
>
> $$=\arg\min_{f(x)}\mathbb{E}_{x_0,x_T}|f(x)-E^{x_0,x_T}(x)|_2^2$$
>
> This makes text-to-image training feasible for drifting models.
>
> **W2: Practical method proposed in Section 4 is modest and its gains are limited.**
>
> The goal of this section is to show that, although IFM-derived techniques are promising, they remain difficult to use in high dimensions because constructing a low-variance multi-sample target remains challenging, as does the corresponding Gini-based analysis.
>
> To better highlight these challenges, we provide additional evaluation on the higher-dimensional FFHQ64 dataset and a Gini analysis across datasets of different dimensionalities (see [*, $\S$1 and $\S$3] for details). The results suggest that the difficulty grows with dimension: FFHQ 64×64 shows less benefit than CIFAR 32×32, and the weights become more degenerate as dimensionality increases.
>
> That said, we agree that we should also demonstrate the potential practical benefits of these techniques. To this end, we add two 2D toy experiments. First, we show the importance of theory-induced volume coverage for IFM in [\*, $\S$5]. Second, we show that the multi-sample target has lower variance and enables faster convergence in [\*, $\S$6].
>
> **W3: Potential benefits of IFM expressivity.**
>
> EFM has an expressive advantage that cannot be captured by CFM. As shown in Table 2 and Figure 2, the two-sided linear interpolant does not benefit from the multi-sample target. By contrast, the electrostatic field satisfies a superposition property, so the objective can be decomposed into one-sided fields induced by the source and target datasets:
>
> $$E(x) = arg\min_{f(x)} | f(x) - \mathbb{E}_{x_0, x_T} E^{x_0, x_T}(x) |_2^2$$
>
> $$=arg\min_{f(x)} |f(x) - \mathbb{E}_{x_0, x_T} ( E^{x_0}(x) - E^{x_T}(x) ) |_2^2$$
>
> $$=arg\min_{f(x)} \mathbb{E}_{x_0} | f(x) - E^{x_0}(x) |_2^2$$
>
> $$+ \mathbb{E}_{x\_T} | f(x) + E^{x\_T}(x) |_2^2$$
>
> Thus, the two-sided multi-sample target reduces to two one-sided targets, which are shown to work better.
>
> **Conclusion remarks:** We hope these clarifications have fully addressed your concerns. We will add considered experiments and discussions in final version. Should any points require further elaboration, we are happy to provide it. Based on these comprehensive explanations, we are hopeful that you will consider raising your rating.
>
> [1] Gen. Modeling via Drifting
> [*] https://anonym.link/?https://zenodo.org/records/19347539

---

> > ### Author Rebuttal · Reviewer_NiHa · 2026-04-02
> >
> > Thank the authors for their detailed response. The rebuttal usefully clarifies the main theoretical claims. It addresses my request for more direct empirical verification of the CFM/forward-only IFM duality on a toy problem, which strengthens the paper. The additional discussion improves the presentation and scope, thus I am raising my score to a 4.

---

> > > ### Author Response · Authors · 2026-04-02
> > >
> > > Thank you very much for your careful and constructive review, and for taking the time to engage with our rebuttal. We are glad that the additional clarifications and toy empirical verification of the duality addressed your concerns, and your comments helped us improve the paper; we will reflect these improvements clearly in the final version.

---

### Decision · Program_Chairs · 2026-04-30

**Decision:**

Reject

**Comment:**

**Summary**

In this paper, the authors first introduce Conditional Flow Matching (CFM) and Interaction Field Matching (IFM). While CFM are an extremely popular formulation of diffusion models, and therefore widely used in practice, IFM are less common. Notable previous works include  EFM (Electrostatic Field Matching). A related construction is PFGM (Poisson Flow Generative Models). The authors show that under certain conditions, namely that the IFM is forward-only, then CFM and IFM are equivalent, thereby resolving the apparent distinction between those two formulations. Despite its theoretical interest, this observation has limited practical impact. The authors turn to the use of multi-sample targets for the velocity estimator, similar to [1]. They highlight a main limitation of this approach which is that the weights concentrate, thereby nullifying the potential benefits of the approach.

**Reviewer concerns**

As reviewer NiHa puts it "The paper’s novelty is strongest at the theorem/unification level rather than at the algorithmic level.

While the constructive equivalence appears nontrivial, the practical method proposed in Section 4 is modest and its gains are limited." Most of the reviewers highlighted the limited impact of the practical findings.


[1] Xu et al. (2021) -- Stable Target Field for Reduced Variance Score Estimation in Diffusion Models